# Active View Selection for Scene-level Multi-view Crowd Counting and Localization with Limited Labels

## Abstract

Multi-view crowd counting and localization fuse the input multi-views for estimating the crowd number or locations on the ground. Existing methods mainly focus on accurately predicting on the crowd shown in the input views, which neglects the problem of choosing the 'best' camera views to perceive all crowds well in the scene. Besides, existing view selection methods require massive labeled views and images, and lack the ability for cross-scene settings, reducing their application scenarios. Thus, in this paper, we study the view selection issue for better scene-level multi-view crowd counting and localization results with cross-scene ability and limited label demand, instead of input-view-level results. We first propose a baseline view selection method (IVS) that considers view and scene geometries in the view selection strategy and conducts the view selection, labeling, and downstream tasks independently. Based on IVS, we put forward an active view selection method (AVS) that jointly optimizes the view selection, labeling, and downstream tasks. In AVS, we actively select the labeled views and consider both the view/scene geometries and the predictions of the downstream task models in the view selection process. Experiments on multi-view counting and localization tasks demonstrate the cross-scene and the limited label demand advantages of the proposed active view selection method (AVS), outperforming existing methods and with wider application scenarios.

## 1 Introduction

Multi-view crowd counting and localization leverage multiple views to predict the crowd count or locations on the ground, alleviating the issue of severe occlusions in large, wide scenes. However, existing multi-view crowd counting and localization methods mainly focus on designing models for accurate estimation of the crowd covered by a randomly selected set of input views (Zhang et al., 2021b), which may not contain or perceive all the crowds well in the scene, resulting in an incorrect prediction of the crowd in the scene. As in Figure 1 top, these frameworks are trained and tested using the ground truth (GT) constructed from the randomly selected views, *i.e.*, not tested on the whole scene.

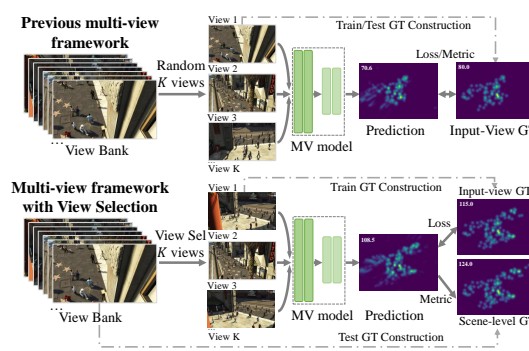

Figure 1: The comparison of existing multi-view counting/localization frameworks and the scene-level multi-view framework with view selection.

Thus, for a complete multi-view vision system, we not only need to design better downstream task models (*e.g.* counting, localization) but also select the best views for better scene-level downstream task performance. A simple two-stage solution is to conduct the view selection first, label the selected views, and then train the downstream multi-view models based on the labeled views, called independent view selection. As shown in Figure 1 bottom, the view-selection-based multi-view framework is trained with GT constructed with selected views, and tested with the scene-level GT,

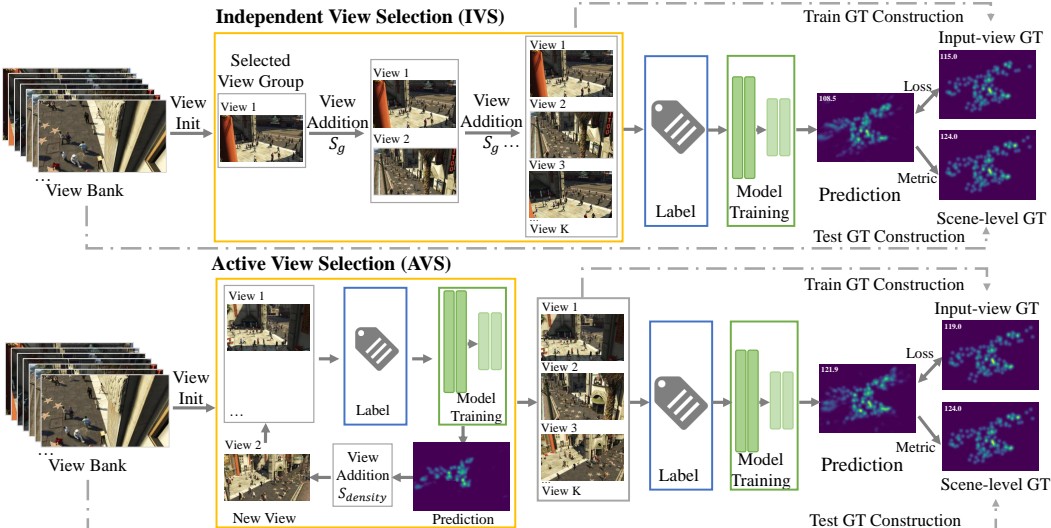

Figure 2: The pipeline of the proposed independent view selection baseline (IVS) and active view selection framework (AVS). IVS (top) separates the view selection (selecting views one by one with view selection score $S_g$ based on scene/view geometries) and downstream model training, while AVS (bottom) jointly conducts view selection and downstream model training: In the view selection process, the downstream model's prediction together and scene/view geometries are both used in view selection score $S_{density}$ to select new views, then the downstream model is trained with the updated view group, repeating the process until finishing selecting $K$ views, finally the downstream model is trained again with the selected and labeled views.

where *the GT constructed with selected views should be close to the scene-level GT*. Recent path planning methods (Zhang et al., 2021a; Liu et al., 2021) adopted the scene/view geometries in view selection for better 3D reconstruction.

Unfortunately, few research works have studied the view selection issue in multi-view counting and localization for whole scene performance. In addition, the two-stage solution divides the view selection and downstream task model training into 2 separate stages, where the priors used for view selection may not be optimal settings for the downstream tasks. Furthermore, to reduce the annotation effort, sometimes only limited views are labeled, causing extra difficulties for downstream model learning. A recent method MVSelect (Hou et al., 2024) proposed a reinforcement learning (RL) framework for view selection and multi-view tasks. However, MVSelect requires GT annotations of *all views*, making it impractical for selecting among large numbers of views to save annotation budget, and has a weak generalization ability due to the RL framework, *making it not applicable to novel scenes*.

To address the mentioned issues, in this paper, we first propose an independent view selection **baseline** (IVS), based on which we further put forward an active view selection framework (AVS), requiring only limited labeled views and with cross-scene abilities. IVS proposes a view selection score equation $S_g$ based on view and scene geometries, including 3 terms: *the scene coverage, the average person-to-camera distance, and the view diversity*, working together to cover the crowd in the scene mostly and clearly. As in Figure 2 top, we first conduct the view selection to expand the selected view group from 1 view to $K$ views according to the proposed view selection score equation $S_g$. Then, we label the selected views and train the multi-view task model on the labeled data. The training and testing GT are constructed from selected input views and all views, respectively.

Furthermore, in contrast to optimizing the view selection and the downstream model independently as in IVS, AVS jointly optimizes the view selection and the downstream tasks by introducing the downstream task predictions in the view selection process. As shown in Figure 2 bottom, the view selection score $S_{denisty}$ considers both the view/scene geometries (similar in IVS) and the prediction results of the downstream task models when expanding the selected view group from 1 to $K$ views, after which the downstream model is trained with the labeled selected views. Thus, the view selection, the data labeling, and the downstream model training are jointly conducted. After $K$ views are reached, the downstream model is trained again with the labeled views. Besides, to reduce

the labeling demand, novel pseudo labels are proposed and utilized to train the downstream models during both the view selection and downstream task model training stages. *The contributions of this paper are*:

- Few research works have studied the view selection problem for scene-level multi-view crowd counting and localization. We propose a novel independent view selection baseline IVS for scene-level downstream tasks, utilizing view and scene geometries in the view selection.
- Based on IVS, we propose an active framework AVS, where the view selection step and the downstream task models are jointly optimized, with better performance than IVS. And we only require limited view labels from the selected views, via adopting pseudo labels on candidate views in the training.
- Our method can apply to novel new scenes, with wider application scenarios, and outperform comparison methods on both multi-view counting and localization.

## 2 RELATED WORK

**Multi-view crowd counting.** Compared to single-image counting (Liang et al., 2022; Han et al., 2023; Zhao et al., 2020; Savner & Kanhangad, 2023; Zhang et al., 2024), multi-view crowd counting is proposed to handle scenes with large areas, severe occlusions, or irregular shapes by fusing multiple synchronized and calibrated camera views. Traditional methods (Ryan et al., 2014; Tang et al., 2015) employed foreground extraction techniques and hand-crafted features, with limited performance and generalization abilities. Recently, deep learning methods (Zhang & Chan, 2019; Zheng et al., 2021; Zhai et al., 2022) have been introduced, trained and tested with ground-plane density maps based on the crowds appearing in the input camera views. CVCS (Zhang et al., 2021b) proposed a camera selection model for cross-view cross-scene multi-view counting with a large synthetic cross-scene multi-view dataset. (Mo et al., 2025) put forward a transformer model with attention-mechanism-based 2D-3D feature lifting. Overall, existing multi-view counting methods focus on the accurate crowd number estimation of the people contained in a randomly selected set of input camera views. *Current SOTAs have not yet explored the problem of selecting the best views for scene-level multi-view counting.* Moreover, existing pretrained single-image models can serve as backbones for multi-view models, thereby improving training speed and effectiveness. And it can directly help us obtain crowd information to enhance perception, such as the frame initialization process of the proposed methods.

**Multi-view crowd localization.** Multi-view crowd localization estimates the crowd locations on the ground in the scene. Early methods' performance is limited (Chavdarova & Fleuret, 2017; Baque et al., 2017) due to no view feature alignment. Recent methods (Song et al., 2021; Hou & Zheng, 2021; Qiu et al., 2022; Liu et al., 2024; Aung et al., 2025) put forward end-to-end frameworks with better performance. MVDet (Hou et al., 2020) used feature perspective transformations to fuse multi-views. CaMuViD (Daryani et al., 2025) facilitates flexible transformation and improves feature fusing across views, removing the need of BEV representation and achieving better detection accuracy. Similarly, *most SOTA multi-view crowd localization methods also focus on estimating crowds 'seen' in the input camera views, not targeting all crowds in the scene.* MVSelect (Hou et al., 2024) is the most related to our paper, and it proposed a reinforcement learning (RL) framework for view selection and downstream tasks. However, MVSelect requires annotations of all views to train the model, and has a weak generalization ability to apply to novel scenes. *In contrast, our method only needs to label the selected views, and with the aid of the proposed pseudo labels, it can be well applied to novel news scenes in the test stage* (see experiments).

**View selection for other multi-view tasks.** View selection is also vital in many other multi-view vision tasks (Majumder et al., 2025; Di Giammarino et al., 2025; Kiciroglu et al., 2020; Border et al., 2018; Zheng et al., 2024; Sun et al., 2021; Ruan et al., 2023; Du et al., 2023), such as path planning, or multi-view object classification. (Liu et al., 2022) measured the reconstructability in a learning way and designed an interactive path planning framework for view selection. MVTN (Hamdi et al., 2021) directly regresses optimal viewpoints for 3D shape recognition with an MLP. (Du et al., 2023) proposed a reinforcement learning-based framework for multi-view active fine-grained visual recognition. (Xiao et al., 2024) proposed a unified framework for view selection methods and devised a thorough benchmark to assess its impact on neural rendering. *It is a trend in other multi-*

Table 1: Summary of main notations and page numbers (P).

| Symbol | Meaning | P | Symbol | Meaning | P |
|---|---|---|---|---|---|
| $F$ | The number of selected frames | 4 | $H_i$ | Visible scene region by view $i$ | 5 |
| $K$ | The number of selected views | 2 | $h$ | Scene height | 5 |
| $v_{max}$ | The view with the largest FOV | 4 | $w$ | Scene width | 5 |
| $V_{select}$ | Selected view group | 4 | $D_p$ | Sum of inverse distance | 6 |
| $S$ | View selection score equation | 16 | $S_{sc}$ | $S$ with scene coverage | 5 |
| $S_g$ | $S$ with view/scene geometries | 5 | $S_{ad}$ | $S$ with average distance | 5 |
| $S_{mask}$ | Mask-indicated $S$ | 6 | $S_{vd}$ | $S$ with view diversity | 5 |
| $S_{density}$ | Density-indicated $S$ | 6 | $B_k$ | Crowd density map mask | 6 |
| $N$ | Downstream task model | 16 | $M_k$ | Crowd density map | 6 |
| $\emptyset$ | No $N$ | 16 | $D_p^{den}$ | $D_p$ with $M_k$ | 6 |
| $H_s$ | Scene region | 5 | $V_{select}^k$ | $k$ selected views | 16 |
| $H_v^k$ | Visible scene region by $V_{select}$ | 5 | $V^g$ | View set of scene's $g$ | 16 |
| $G$ | Scene set | 16 | $E$ | The number of training epoch | 16 |
| $V_{select}^g$ | Selected views of scene $g$ | 16 | | | |

*view tasks to jointly conduct the view selection and the downstream task. However, there is little research on view selection for scene-level multi-view crowd counting and localization tasks*, which is a relatively unexplored area.

## 3 ACTIVE VIEW SELECTION FRAMEWORK

We first propose a novel independent view selection **baseline** (IVS) adopting a two-stage process for scene-level downstream tasks. Next, based on IVS, we propose the active view selection framework (AVS) for jointly optimizing view selection and downstream task model training. Pseudo labels are adopted to enhance the model's cross-scene generalization abilities. For both IVS and AVS, we assume an annotation budget of $F$ frames per view and $K$ views of each scene, or a total $FK$ images per scene. We provide a summary table of the main notations with page numbers (P) shown in Table 1 to facilitate symbol querying.

### 3.1 INDEPENDENT VIEW SELECTION BASELINE (IVS)

In IVS, we first initialize the selected frames and the first view, then start to add new views one by one with the proposed view selection score equation based on scene/view geometries, expanding the selected view number from 1 to $K$.

#### 3.1.1 INITIALIZATION

Initialization has two stages: selecting the $F$ frames to be processed and selecting the first view. For frame selection, we first find the view $v_{max}$ with the largest field-of-view (FOV) area on the ground. Then, we select the first frame as the one with the largest predicted crowd count in view $v_{max}$ using DM-Count (Wang et al., 2020a), a pre-trained single-image counting model that can help effectively perceive crowd information in a label-efficient manner. Next, given the diversity in the limited labels, we select the rest frames with the lowest cosine similarity between the selected frames and candidate frames of view $v_{max}$. This process is repeated until $F$ frames are selected. For view initialization, we select the view with the largest FOV as the first view in the selected view group (denoted as $V_{select}$) in IVS, and the view with the largest crowd count sum across all selected $F$ frames is selected as the first view in AVS.

#### 3.1.2 VIEW ADDITION: $S_g$

With the selected frames generated by frame initialization and the selected view group $V_{select}$ including the first view from view initialization above, a view selection score equation $S_g$ is proposed for view addition. For each iteration, the $S_g$ score of each candidate view together with the current $V_{select}$ is calculated, and then we select the new view with the largest score. The process is repeated until the specified $K$ views are selected (Please refer to Algorithm 3 in the Appendix for the de-

tails). $S_g$ consists of 3 terms: Scene Coverage, Average Distance, and View Diversity, which are as follows.

**Scene Coverage** score term $S_{sc}$ indicates whether the selected views can cover all crowds in the scene as much as possible. We first use the scene ground plane map as the scene region $H_s$, whose area size is $Area(H_s) = hw$, and $h$ and $w$ are the height and width of the map. Then, we calculate the visible scene areas covered by the selected views as $H_v^k = \{H_1 \cup H_2...H_k\}$, which is the combined FOV region of $k$ views (see Figure 3 (a) and (b)), and $H_i$ is view $i$'s FOV covering region. Thus, the area ratio of the visible region $H_v^k$ and the scene region $H_s$ is defined as $S_{sc}$:

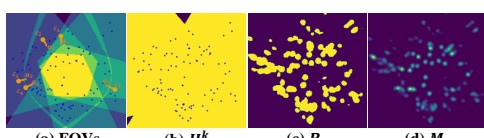

Figure 3: (a) The combined FOVs of selected views; (b) The FOV mask; (c) The crowd mask; (d) The crowd density. The dots in (a) and (b) are ground-truth crowd locations.

$$S_{sc} = \sum H_v^k / Area(H_s) = \sum \{H_1 \cup H_2...H_k\}/(hw), \tag{1}$$

where a larger $S_{sc}$ indicates a higher probability of covering all crowds by the selected views.

**Average Distance** score term $S_{ad}$ considers the average person-to-camera distance in the view selection, indicating whether the selected views can 'see' the crowd clearly. Specifically, for each location $p$ in the visible region $H_v^k$, the person's inverse distance to the currently selected $k$ cameras is calculated as $D_p = \sum_{i=1}^{k} 1/\|p - c_i\|$, where $c_i$ is the $i$-th camera's location on the ground (see Figure 3 (a)) and $p$ is in $H_i$, where higher values indicates shorter distance to the selected cameras. Combining all crowds, $S_{ad}$ is:

$$S_{ad} = \sum_{p \in H_v^k} D_p / \sum H_v^k. \tag{2}$$

As the crowd locations are not known, all locations in the approximate visible region $H_v^k$ are used in the calculation. Similarly, a higher $S_{ad}$ forces the selected cameras to be close to the crowds and captures the crowds more clearly.

**View Diversity** score term $S_{vd}$ avoids the setting that all cameras are located at the same place and point out. Because we require multi-cameras to be placed at different corners facing each other to make full use of their occlusion handling potential (Zhang & Chan, 2019). Thus, we adopt a similarity measure (Zhou et al., 2020) to calculate $S_{vd}$:

$$S_{vd} = \exp\left(-\lambda \sum_{i=1}^{k} \sum_{j=i+1}^{k} \frac{\mathbf{o}_i \cdot \mathbf{o}_j}{\|c_i - c_j\| + \epsilon}\right), \tag{3}$$

where $\mathbf{o}_i, \mathbf{o}_j$ are the camera optical axis directions (see Figure 3a), $c_i, c_j$ are camera locations, $\lambda$ is a hypeparameter, and $\epsilon$ is a small value to avoid zero denominator. When the selected cameras have larger view direction and location differences, i.e., view diversity, $S_{vd}$ is higher.

By combining the 3 terms, we obtain the independent view selection score equation $S_g$ by only considering the view and scene geometries.

$$S_g = S_{sc} * S_{ad} * S_{vd} \tag{4}$$

$$= \frac{\sum_{p \in H_v^k} D_p}{Area(H_s)} \exp\left(-\lambda \sum_{i=1}^{k} \sum_{j=i+1}^{k} \frac{\mathbf{o}_i \cdot \mathbf{o}_j}{\|c_i - c_j\| + \epsilon}\right). \tag{5}$$

Once the required frames and views are selected, they are annotated, and then the downstream task model is trained on the labeled data (see Figure 2 top right). The main **weakness** of the independent view selection baseline (IVS) is that the view selection and downstream model training are separated, which does not ensure an optimal result for scene-level tasks. For example, the selected views are not necessarily suitable for downstream model training due to multi-view counting and localization models being sensitive to view angles, heights, or other properties.

## 3.2 ACTIVE VIEW SELECTION (AVS)

To address the weakness of the IVS baseline–optimizing the view selection and downstream model separately, we propose the AVS framework that jointly optimizes the view selection and downstream task models as in Figure 2 bottom. In the view selection process, the intermediate model's prediction together with the scene/view geometries are adopted in the view selection score $S_{density}$, and

selected new views are labeled and used to train the downstream model, which is repeated until the desired view number is reached. Finally, the downstream model is trained again with the selected and labeled views. The complete algorithm procedure is presented in Algorithm 2 of the Appendix. We update the view selection score equation in IVS by introducing the downstream task predictions, denoted as $S_{mask}$ and $S_{density}$, with details as follows.

**Mask-indicated view selection** $S_{mask}$**.** The visible region in (5) is defined by the combination of the FOVs of the selected cameras, which neglects the actual crowd regions in the scene. Therefore, we rely on the prediction density maps $M_k$ from the downstream model $N$ of the selected $k$ views $\{v_1, v_2, ..., v_k\}$ to more accurately indicate the appearing crowds' regions in the scenes: $M_k = N(v_1, v_2, ..., v_k)$. Specifically, we binarize $M_k$ with a threshold $\sigma$ to obtain a crowd density map mask $B_k$ (see Figure 3c), which is used to replace $H_v^k$ in (1), (2) and (5). Thus, we obtain

$$S_{mask} = \frac{\sum_{p \in B_k} D_p}{Area(H_s)} \exp\left(-\lambda \sum_{i=1}^{k} \sum_{j=i+1}^{k} \frac{\mathbf{o}_i \cdot \mathbf{o}_j}{\|c_i - c_j\| + \epsilon}\right). \tag{6}$$

where $B_k$ indicates *the crowd location information, which is utilized in the view selection process*. Note that the downstream counting or localization model $N$ is involved in the view selection score term $S_{mask}$, while the newly selected views during the view selection process could be fed into and train the downstream model. Therefore, the view selection and downstream task model are interacting in these two steps and thus influence each other.

**Density-indicated view selection** $S_{density}$**.** The mask-indicated view selection uses the binarized density map $B_k$ to indicate the crowd-visible regions, which neglects the crowd density's influence on the view selection score term. In other words, the crowded areas with higher densities should have higher weights in the view selection process. Therefore, we propose the density-indicated view selection score $S_{density}$ by introducing the density prediction $M_k$ (see Figure 3d) of the downstream task model into $D_p$ in (2), which is rewritten as $D_p^{den} = \sum_i^k \frac{M_k(p)}{\|p - c_i\|}$, where $M_k(p)$ indicates the density value of point $p$. Thus, *by updating $S_{ad}$ with $D_p^{den}$, and replacing $H_v^k$ with $B_k$ in $S_{sc}$ and $S_{ad}$*, the view selection score term in (6), we obtain:

$$S_{density} = \frac{\sum_{p \in B_k} D_p^{den}}{Area(H_s)} \exp\left(-\lambda \sum_{i=1}^{k} \sum_{j=i+1}^{k} \frac{\mathbf{o}_i \cdot \mathbf{o}_j}{\|c_i - c_j\| + \epsilon}\right). \tag{7}$$

The view selection score term $S_{density}$ considers both *the view/scene geometries, and the crowds' density level and location information*, where the view selection and downstream model training are conducted jointly.

**Pseudo labels and training.** To enhance the model's cross-scene generalization ability, we utilize novel pseudo labels (see more details in Appendix) to better train the downstream model. During the view selection, the currently selected views $V_{select}^k$ and a random unselected view are combined as pseudo inputs to train the model, whose GT is ground-plane density maps of crowds covered by $V_{select}^k$ and masked by $V_{select}^k$'s combined FOV masks ($H_v^k$) in the loss. Besides, after the view selection, the selected views $V_{select}^K$ of the $F$ selected frames are used for downstream model training. In addition to that, we also add pseudo inputs in training, which is a mix of 1 selected view and $K-1$ unselected random views, whose pseudo-GT is the $K$ selected views' ground truth ground plane density maps and masked by the intersection of $H_v^K$ and the pseudo input views' combined FOV mask in the loss. The ratio of the two kinds of multi-view inputs is 1:1. By using pseudo labels, a large number of unlabeled views are included in the model training, significantly improving the model's generalization abilities. *Both IVS and AVS adopted pseudo labels in the model training.*

# 4 EXPERIMENTS AND RESULTS

## 4.1 MULTI-VIEW CROWD COUNTING

**Experiment design.** In the training, IVS conducts the view selection, labeling, and downstream model training independently, while AVS conducts them jointly until the view number reaches $K$ and then trains the downstream task model on the labeled $K$ views. In the testing, no model training is needed for IVS or AVS, where the same view selection process is conducted with all testing frames, and the downstream model prediction is directly used in the view selection score.

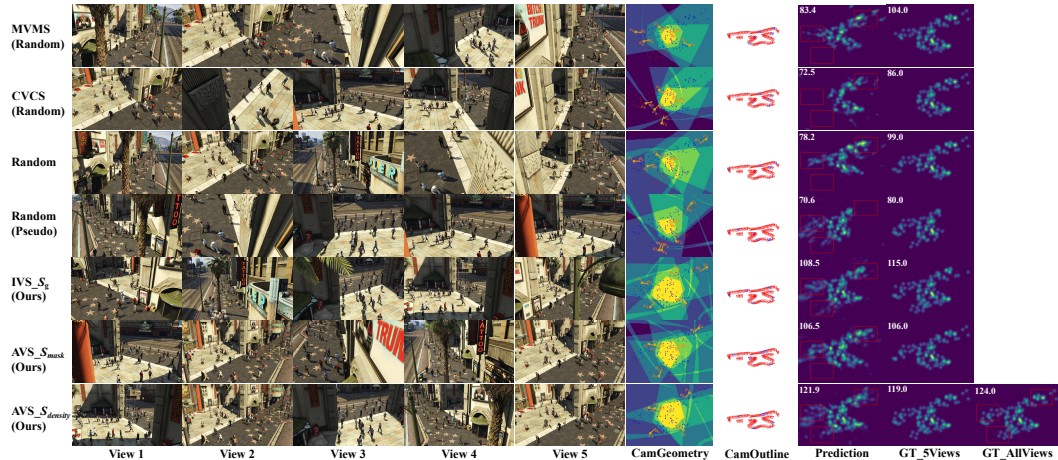

Figure 4: The view selection and multi-view counting results on CVCS: our methods select better views covering the whole scene and predict better density maps close to the scene-level crowd GT (GT_AllViews). Blue camera view indicates the selected view in column CamOutline.

Table 2: Comparison of the multi-view counting results on the CVCS dataset.

| Method | MAE ↓ | MSE ↓ | NAE ↓ | CoverRate ↑ |
|---|---|---|---|---|
| MVMS (Random) | 36.65 | 43.03 | 0.271 | 0.885 |
| CVCS (Random) | 39.18 | 44.92 | 0.289 | 0.885 |
| Uniform | 21.76 | 25.75 | 0.163 | 0.945 |
| Random | 36.59 | 42.06 | 0.271 | 0.885 |
| Random (Pseudo) | 28.22 | 33.73 | 0.208 | 0.885 |
| Random (Oracle) | 15.37 | 20.91 | 0.115 | 0.885 |
| IVS_$S_g$ (Baseline, Ours) | 14.98 | 18.93 | 0.111 | 0.959 |
| AVS_$S_{mask}$ (Ours) | 12.53 | 15.33 | 0.093 | 0.955 |
| AVS_$S_{density}$ (Ours) | **10.99** | **13.57** | **0.083** | **0.960** |

**Dataset.** The multi-view counting task is conducted on a multi-scene dataset CVCS (Zhang et al., 2021b), training on 23 scenes and testing on 8 scenes, with 60-120 camera views in each scene with calibrations, which is challenging and suitable to *validate the cross-scene generalization of the proposed frameworks*. We only require $F = 20$ frames and $K = 5$ views are labeled in each scene. We also conduct experiments on a real dataset a real dataset CityStreet (Zhang & Chan, 2019). CityStreet contains 3 camera views and 300 frames for training and 200 for testing. We use $F = 60$ frames and $K = 2$ views for task settings.

**Comparison methods.** We compare the proposed AVS with the IVS baseline, the random view selection methods 'Random', 'Random (Pseudo)', and 'Random (Oracle)'. 'Random' randomly selects 5 views at once, and then trains on the selected views. 'Uniform' uses the same multi-view counting model as ours, but replaces the view selection method with the uniform view sampling from all views. 'Random (Pseudo)' adds views one-by-one as in AVS but in a random way, and also adopts pseduo-label training. Both share the same multi-view counting model architecture as ours. 'Random (Oracle)' randomly selects 5 views at once and uses the selected 5-view GT as a prediction (the 'best' counting model). We also compare with previous SOTAs CVCS and MVMS with the same labeling budget using the random selection way, denoted as 'CVCS (Random)' and 'MVMS (Random)'.

**Implementation details.** The input image resolution is 640x360, and a random 160x180 cropping strategy on the scene map is adopted in the training. For AVS, during each view expansion iteration, an MAE threshold $\tau$ of 20 is adopted to stop the multi-view counting model training for the next view addition. We use the backbone model in CVCS method with a feature pyramid fusion net (FPN) as the downstream multi-view counting model. The model is trained using the SGD optimizer with a learning rate of 1e-3. $\epsilon$ is 1e-10 and $\lambda$ is 0.1 in (3), and threshold $\sigma$ is the mean of density map $M_k$.

**Evaluation metrics.** We use mean absolute error (MAE), root mean squared error (MSE), and normalized absolute error (NAE) of the predicted crowd count and the *scene-level* ground-truth

Table 3: The ablation study on the terms of the AVS score equation $S_{density}$ on CVCS dataset.

| Term | MAE↓ | MSE↓ | NAE↓ |
|---|---|---|---|
| $S_{sc}$ | 16.44 | 21.01 | 0.125 |
| $S_{sc} * S_{ad}$ | 18.56 | 23.39 | 0.138 |
| $S_{sc} * S_{vd}$ | 14.77 | 18.31 | 0.108 |
| All (Ours) | **10.99** | **13.57** | **0.083** |

Table 4: Comparison of the multi-view counting results on the CityStreet dataset.

| Method | MAE↓ | MSE↓ | NAE↓ |
|---|---|---|---|
| Random | 13.47 | 16.60 | 0.170 |
| Uniform | 11.82 | 15.00 | 0.130 |
| IVS_$S_g$ | 11.28 | 14.36 | 0.128 |
| AVS_$S_{density}$ | **9.80** | **11.93** | **0.118** |

count (all crowds in the scene) as counting metrics. Besides, we also use the percentage of the crowds covered by the selected views among all crowds in the scene to evaluate different view selection methods, denoted as 'CoverRate'. Thus, the metrics not only assess the counting model's performance but also reflect whether the selected views can adequately cover all crowds.

**Multi-view crowd counting results.** We compare the proposed AVS with the IVS baseline and other comparison methods in Table 2 on the CVCS dataset – AVS achieves the best performance. 'MVMS (Random)' and 'CVCS (Random)' achieve much worse results because they input random views without good view selection for scene-level counting. We are also better than 'Random', 'Random (Pseudo)', and 'Random (Oracle)', demonstrating the advantage of the proposed view selection frameworks over random selection strategies, even with the selected view GT as predictions (the best counting model). Compared to 'Random' view selection, 'Uniform' achieves better performance. But our methods consider view/scene geometries and interaction with downstream models, achieving the best results. **Compared to IVS**, AVS is much better, either with $S_{mask}$ or $S_{density}$. Even though IVS_$S_g$ has a close CoverRate to AVS_$S_{density}$, its scene-level counting performance is much worse than AVS_$S_{density}$. This shows the superiority of AVS, which optimizes the view selection and the multi-view counting model training jointly for better scene-level results. $S_{density}$ is better than $S_{mask}$ because $S_{density}$ considers the crowd density levels as well as the location information in view selection, while $S_{mask}$ only utilizes the location information. Note that the CVCS dataset is a multi-scene dataset, and our methods could perform *cross-scene training and testing*, demonstrating the flexibility and generalization ability. We also compare the proposed view selection methods with other comparison methods on CityStreet in Table 4. The proposed methods achieve the best results, further indicating that equipping a well-designed view selection method is essential for scene-level tasks with limited labels.

The **visualizations** on CVCS are shown in Figure 4, where the inputs are variable for different methods. 'GT_5Views' and 'GT_AllViews' are the GT constructed from the 5 selected views or all views. The former is used for training and the latter is for evaluation. It's observed that the 'GT_5Views' of our method 'AVS_$S_{density}$' contains the most crowds, and our method can also cover more crowds in the scene (red dots in 'CamGeometry' indicate crowds not covered by the selected views), indicating the efficacy of our view selection method. The predictions also demonstrate our method's advantages, while comparison methods neglect the regions highlighted by red boxes.

**View selection terms ablation study.** We perform ablation studies on the usage of the 3 terms in $S_{density}$ in Table 3: using $S_{sc}$, using $S_{sc} * S_{ad}$ or $S_{sc} * S_{vd}$, or using all 3 terms (namely $S_{density}$). Compared to only using $S_{sc}$, adding $S_{vd}$ improves the results, while adding $S_{ad}$ without the view diversity term $S_{vd}$ achieves worse results, due to selected views being placed at the similar locations and directions, reducing the multi-view fusion performance (as shown in Figure 5b). Using all 3 terms

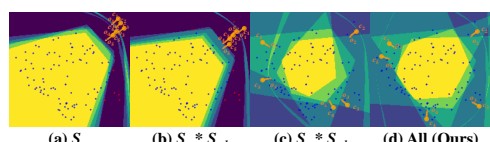

(a) $S_{sc}$    (b) $S_{sc} * S_{ad}$    (c) $S_{sc} * S_{vd}$    (d) **All (Ours)**

Figure 5: The selected view positions ($c_j$) and directions ($o_j$) for different terms. Red dots are uncovered crowds.

is the best because it can select views covering most of the crowds with a larger overlapping area for better multi-view fusion (see Figure 5d), which indicates each term's contribution to the final view selection performance.

**View and frame number ablation study.** We conduct ablation studies on the selected view number $K$ and frame number $F$ for AVS_$S_{density}$ in Table 5, with other settings kept the same (except $\tau = 30$ for 5 frames for its poor performance). As $K$ increases, more views are provided to cover the whole scene, generally achieving better scene-level counting performance. With more frames, the multi-view counting model is trained with more labeled data, achieving better results, too.

Table 5: The ablation study on the selected view/frame number $K$ (keep $F = 20$)/$F$ (keep $K = 5$) on CVCS dataset.

| $K$ | MAE | MSE | NAE | $F$ | MAE | MSE | NAE |
|---|---|---|---|---|---|---|---|
| 3 | 15.95 | 20.13 | 0.118 | 5 | 19.01 | 22.91 | 0.144 |
| 5 | 10.99 | 13.57 | 0.083 | 10 | 11.69 | 14.56 | 0.088 |
| 7 | 10.57 | 13.04 | 0.079 | 20 | 10.99 | 13.57 | 0.083 |
| 9 | **9.82** | **12.24** | **0.072** | 40 | **10.31** | **12.87** | **0.077** |

Table 6: The ablation study on when to add pseudo label training for AVS$\_S_{density}$ (Ours) on CVCS dataset.

| Pseudo | MAE $\downarrow$ | MSE $\downarrow$ | NAE $\downarrow$ |
|---|---|---|---|
| None | 20.17 | 24.77 | 0.156 |
| ViewSel | 19.89 | 24.62 | 0.154 |
| ModelTrain | 11.32 | 14.69 | 0.083 |
| Both (Ours) | **10.99** | **13.57** | **0.083** |

Table 7: Comparison of the multi-view localization results on Wildtrack and MultiviewX. AVS achieves the best performance among all partial-labeled methods (3 views) and outperforms MVSelect trained with the ground truth of full labels (all views). Bold indicates the best result among all partial-labeled methods. We also achieve close performance to other full-labeled SOTAs.

| | Dataset | MultiviewX | | | | | Wildtrack | | | | |
|---|---|---|---|---|---|---|---|---|---|---|---|
| Label | Method | MODA↑ | MODP↑ | P↑ | R↑ | F1↑ | MODA↑ | MODP↑ | P↑ | R↑ | F1↑ |
| Full | MVDet (Hou et al., 2020) | 83.9 | 79.6 | 96.8 | 86.7 | 91.5 | 88.2 | 75.7 | 94.7 | 93.6 | 94.1 |
| | SHOT (Song et al., 2021) | 88.3 | 82.0 | 96.6 | 91.5 | 94.0 | 90.2 | 76.5 | 96.1 | 94.0 | 95.0 |
| | MVDeTr (Hou & Zheng, 2021) | 93.7 | 91.3 | 99.5 | 94.2 | 97.8 | 91.5 | 82.1 | 97.4 | 94.0 | 95.7 |
| | 3DROM (Qiu et al., 2022) | 95.0 | 84.9 | 99.0 | 96.1 | 97.5 | 93.5 | 75.9 | 97.2 | 96.2 | 96.7 |
| | MVSelect (Hou et al., 2024) | 88.1 | 89.8 | 98.2 | 89.7 | 93.8 | 88.6 | 79.9 | 93.3 | 94.2 | 93.7 |
| Partial | Random | 85.3 | 80.8 | 97.3 | 87.7 | 92.2 | 80.6 | 75.8 | 93.0 | 87.1 | 89.8 |
| | Random (Pseudo) | 85.5 | 81.1 | 97.5 | 87.7 | 92.4 | 82.8 | 75.4 | 93.8 | 88.5 | 91.0 |
| | IVS$\_S_g$ (Ours) | 86.4 | 81.2 | 97.6 | 88.6 | 92.9 | 87.3 | **77.2** | 93.7 | **93.6** | 93.6 |
| | AVS$\_S_{mask}$ (Ours) | 87.9 | 80.5 | 97.3 | 90.4 | 93.7 | 87.7 | 77.0 | 95.5 | 92.0 | 93.7 |
| | AVS$\_S_{density}$ (Ours) | **89.2** | **82.1** | **98.0** | **91.0** | **94.4** | **89.6** | 76.7 | **96.1** | 93.4 | **94.7** |

**Pseudo labels ablation study.** The ablation studies on the pseudo labels for AVS$\_S_{density}$ are shown in Table 6. We compare the method without using the pseudo labels in the model training (None), using pseudo labels only at the view selection stage (ViewSel), using pseudo labels only at the final model training stage when view selection is finished (ModelTrain), or using pseudo labels at both stages (Ours). The results show that pseudo labels can indeed improve the performance when added at any stage, and adding pseudo labels at both stages can obtain the best performance. *See more results in Appendix.*

## 4.2 Multi-view Crowd Localization

**Datasets and comparisons.** The multi-view crowd localization task is evaluated on two single-scene datasets Wildtrack and MultiviewX, where the same settings are used as in MVSelect (Hou et al., 2024): 360 frames are all used for model training and 40 frames are for testing, without frame selection, and only $K = 3$ views are selected and labeled. We compare the proposed active view selection framework, the independent view selection baseline, with the random view selection method in multi-view counting tasks. We also compare with an RL-based view selection comparison method MVSelect with the same multi-view localization model. We also compare with SOTA methods trained with full labels. *Note that MVSelect uses* all *view labels for joint view selection and crowd localization model training, while we only need to label the selected 3 views.*

**Implementation details.** During the view selection process, the multi-view crowd localization model training threshold $\tau$ is MODA=40, namely the model training stops when MODA reaches 40, then we add the next view. Unlike MVSelect, which uses annotations from all views for training, we label only the selected views and apply pseudo-labels to incorporate unlabeled views during training. We use the same MVDet (Hou et al., 2020) implemented in MVSelect as the downstream multi-view localization model, trained with data augmentation and focal loss in MVDet. The model is trained using the SGD optimizer, and the learning rate is 1e-2 and 5e-2 for Wildtrack and MultiviewX, respectively. $\sigma$ is 0.6 in (6), and $\lambda$ and $\epsilon$ are the same as in multi-view counting settings.

**Metrics.** We use Multiple Object Detection Accuracy (MODA), Multiple Object Detection Precision (MODP with distance threshold $t = 0.5m$ in MVDet), Precision (P), Recall (R), and F1_score (F1) as metrics.

**Multi-view crowd localization results.** As shown in Table 7, we compare our methods with other view-selection-based methods (MVSelect, Random, and Random (Pseudo)) and full-label supervised methods. AVS outperforms the random view selection methods 'Random', 'Random

Table 8: The ablation study on the pseudo labels.

| Dataset | MultiviewX | | | | | Wildtrack | | | | |
|---|---|---|---|---|---|---|---|---|---|---|
| Pseudo | MA. | MP. | P | R | F1 | MA. | MP. | P | R | F1 |
| None | 86.2 | 73.0 | **98.4** | 87.6 | 92.7 | 79.6 | **77.6** | 94.2 | 84.9 | 89.3 |
| ViewSel | 86.9 | 79.8 | 97.6 | 89.1 | 93.1 | 83.6 | 75.9 | 94.7 | 88.6 | 91.5 |
| ModelTrain | 87.8 | 81.6 | 98.0 | 89.7 | 93.6 | 79.9 | 77.5 | 95.8 | 83.6 | 89.3 |
| Both | **89.2** | **82.1** | 98.0 | **91.0** | **94.4** | **89.6** | 76.7 | **96.1** | **93.4** | **94.7** |

(Pseudo)', and 'MVSelect', demonstrating its advantages of using joint optimization of the view selection and downstream model training. Compared to IVS, AVS achieves better performance, either with $S_{mask}$ or $S_{density}$. $S_{density}$ is better than $S_{mask}$, which also proves AVS's effectiveness due to considering both crowd density-level information and location information, and the view/scene geometries in the view selection. Compared to MVDet, SHOT, MVDeTr, and 3DROM, which are trained on all input views and labels, the proposed active view selection framework ($S_{density}$) outperforms MVDet on both MultiviewX and Wildtrack, also proving the advantages of our methods. Note that MVSelect also relies on all camera view labels (annotations and calibrations) in the model training and cannot perform on novel new scenes with different view and scene settings, while *our methods only rely on limited view labels with wider application scenarios (as on CVCS)*.

**Pseudo labels ablation study.** The ablation studies on the pseudo labels for the active view selection framework ($S_{density}$) are shown in Table 8: no pseudo labels (None), adding at view selection (ViewSel) or final model training stage (M.Train) after view selection, or both (Ours). Similarly, we add the pseudo-label training at different stages and compare their influence on the performance. The results show that regardless of which stage, pseudo labels can improve the performance. On Wildtrack, adding pseudo labels is more effective at the view selection stage. The possible reason is that the view difference is larger in Wildtrack, and thus the pseudo-label training is more useful for the model to generalize to new views. Anyway, adding pseudo-label training at both stages can achieve the best performance. **See more ablation study results in the Appendix**.

## 5 CONCLUSION

In this paper, we focus on the view selection issue for scene-level multi-view crowd counting and localization tasks. We first propose the independent view selection baseline (IVS) by considering the view and scene geometries. Then, based on IVS, we propose the active view selection method (AVS), which considers the downstream model predictions in the view selection and jointly optimizes the view selection and downstream tasks. Extensive experiments on the two tasks reveal the advantages of the proposed AVS method compared to all comparisons. The proposed method can apply to novel scenes with limited labels, demonstrating its better generalization abilities and wider application scenarios. In the future, the method could also be extended to other BEV-based or 3D reconstruction tasks to reduce labeling costs.

**Ethics statement.** In our work, we use public synthetic and real datasets for designing view selection frameworks for multi-view crowd counting and localization. Our model does not directly rely on human face information, or track people in videos, either. Besides, our paper reduces the demand for head-labeled human images by view selection and using pseudo labels in the model training, which can also reduce privacy concerns. For privacy protection, we could mask out human faces in real datasets or rely on synthetic datasets more for research. Surveillance of large crowds has safety applications, *e.g.*, for detecting overcrowded areas or crowd anomalies. Multi-view crowd counting and localization could be used for crowd analysis, autonomous driving, public traffic management, etc. For applications, we could encode the images as features or mask out human faces first instead of directly inputting images with human faces to the crowd counting and localization models to improve privacy protection.

**Reproducibility statement.** In addition to the illustration of the main text, more details of the proposed method, including the independent view selection and active view selection framework, are introduced in the Appendix A. The complete pipelines for training and testing are presented in the Appendices A.2 and A.2.1, which include visualizations and algorithm procedures. For pseudo-label, the concise visualizations are shown in the Appendix A.2 for understanding the construction of pseudo-label and the corresponding ground-truth easily, achieving more friendly comprehension and reproducibility.

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

# A    EXTRA DETAILS

## A.1    METRICS

**Multi-view crowd counting.** We use mean absolute error (MAE), root mean squared error (MSE), and normalized absolute error (NAE) of the predicted crowd count and the *scene-level* ground-truth count (all crowds in the scene) as metrics:

$$MAE = \frac{1}{N} \sum_{i=1}^{N} |y_i^{gt} - \hat{y}_i|, \tag{8}$$

$$MSE = \sqrt{\frac{1}{N} \sum_{i=1}^{N} |y_i^{gt} - \hat{y}_i|^2}, \tag{9}$$

$$NAE = \frac{1}{N} \sum_{i=1}^{N} \frac{|y_i^{gt} - \hat{y}_i|}{y_i^{gt}}, \tag{10}$$

where $N$ is the number of the samples, and $\hat{y}_i$ and $y_i^{gt}$ are the predicted count and the corresponding ground truth (GT) count of the $i$-th sample, respectively. In evaluation, the GT count refers to the crowd number in all views, not the crowd count of the selected views, to indicate the scene-level counting performance. Thus, the metrics not only assess the performance of the counting model but also reflect whether the selected views can adequately cover all crowds.

The percentage of the crowds covered by the selected views among all crowds in the scene is used to evaluate different view selection methods, and is denoted as 'CoverRate':

$$CoverRate = \frac{1}{N} \sum_{i=1}^{N} \frac{y_i^{gt5}}{y_i^{gt}}, \tag{11}$$

where $y_i^{gt5}$ and $y_i^{gt}$ denote the crowd number in the selected and all views, respectively. 'CoverRate' indicates the crowd coverage performance of the selected views.

Besides, we also evaluate the mean coverage rate of each selected view:

$$CoverRate_{mean} = \frac{1}{N} \sum_{i=1}^{N} \frac{\sum_{j=1}^{K} y_{ij}^{gt}/y_i^{gt}}{K}, \tag{12}$$

where $y_{ij}^{gt}$ and $y_i^{gt}$ denote the number of the crowd covered by the $j$-th selected view and all views in the scene, respectively, and $K$ is the number of the selected views.

**Multi-view crowd localization.** We use Multiple Object Detection Accuracy (MODA), Multiple Object Detection Precision (MODP), Precision (P), Recall (R), and F1_score (F1) as metrics to evaluate the multi-view crowd localization performance. $MODA = 1 - (FP + FN)/(TP + FN)$ measures the overall performance. $MODP = (\sum(1 - d[d < t]/t))/TP$ measures the localization precision, where $d$ is the distance from a detected person point to its ground truth and $t$ is the distance threshold set to 0.5m as in (Hou et al., 2020). $P = TP/(FP + TP)$, $R = TP/(TP + FN)$, and $F1 = 2P * R/(P + R)$. Here, $TP$, $FP$, and $FN$ are the number of true positives, false positives, and false negatives, respectively.

## A.2    IMPLEMENTATION DETAILS

**Initialization details.** For AVS, the initialization consists of two stages: selecting the $F$ multi-frames and selecting the initial view. For **multi-frame selection**, we first find the view $v_{max}$ with the largest field-of-view (FOV) area on the ground. Then, we select the first multi-frame as the one with the largest predicted crowd count in view $v_{max}$ using a pre-trained single-image counting model DM-Count (Wang et al., 2020a) trained on the NWPU dataset (Wang et al., 2020b). Then, we select the next frame by calculating the cosine similarity between the selected first frame and each remaining unselected frame in view $v_{max}$, respectively. The unselected frame with the lowest cosine similarity will be selected as the second frame. If there is more than one selected frame, each

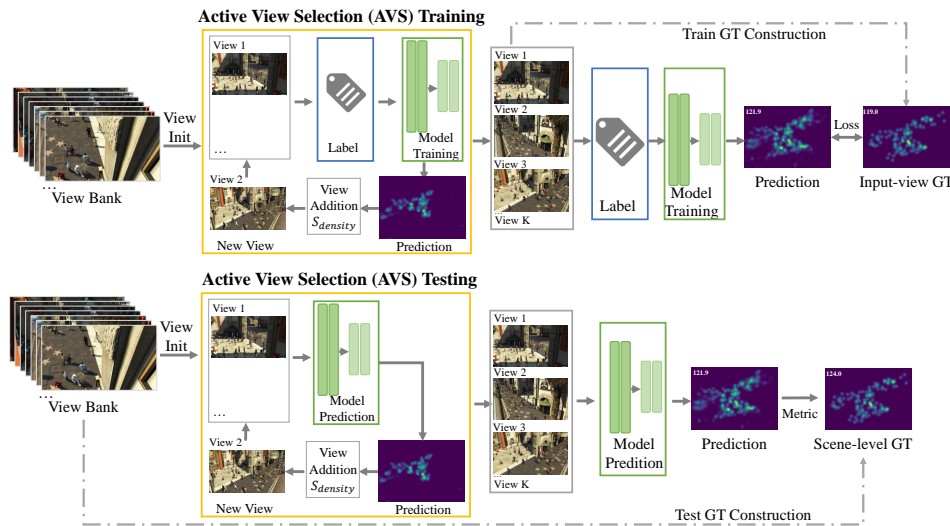

Figure 6: The training and testing pipeline of the proposed independent view selection framework (IVS) for scene-level multi-view crowd counting and localization tasks.

Figure 7: The training and testing pipeline of the proposed active view selection framework (AVS) for scene-level multi-view crowd counting and localization tasks. Top: AVS jointly conducts view selection, view labeling, and downstream model training; Bottom: In the test, AVS uses the prediction result from the downstream task model to conduct view selection without additional model training.

unselected frame will be used to calculate the cosine similarity with all the selected frames in view $v_{max}$. The frame with the lowest cosine similarity sum across all selected frames will be selected. This process is repeated until $F$ multi-frames are selected. Especially, in the test, all the frames from the test set will be used to conduct view selection by default.

For **view initialization**, we select the view with the largest crowd count total across all selected $F$ multi-frames as the first view in the selected view group $V_{select}$. In particular, for multi-view crowd localization tasks, all the frames ($F = 360$) in the training set are used to conduct view selection and train the downstream model. For IVS, whose frame initialization is the same as AVS (described above), the view with the largest FOV area on the ground is selected as the first view since IVS only considers view/scene geometries in the view selection.

---

**Algorithm 1** Independent View Selection Framework

---

1: **Input**: each scene's total views $V^g \in \{v_1^g, ..., v_n^g\}$, all the scenes $G = \{g\}$, max selected view number $K$, view selection score equation $S = S_g$, task model $N$.
2: initialize frames and the selected view group $\{V_{select}^g\}$.
3: **for** $g \in G$ **do**
4:     **for** $k \in \{2, \dots, K\}$ **do**
5:         view_addition($V^g$, $V_{select}^g$, $S$, N);
6:     **end for**
7: **end for**
8: label all the selected views $\{V_{select}^g\}$;
9: model_training($N$, $\{V_{select}^g\}$).

---

---

**Algorithm 2** Active View Selection Framework

---

1: **Input**: each scene's total views $V^g \in \{v_1^g, ..., v_n^g\}$, all the scenes $G = \{g\}$, max selected view number $K$, training epochs $E$, threshold $\tau$ to add view, view selection score equation $S \in \{S_{mask}, S_{density}\}$, and task model $N$.
2: initialize frames and the selected view group $\{V_{select}^g\}$.
3: label all the selected views $\{V_{select}^g\}$.
4: **for** $e \in \{1, \dots, E\}$ **do**
5:     $metric$ = model_training($N$, $\{V_{select}^g\}$);
6:     **if** $metric > \tau$ **and** $len(V_{select}^g) < K$ **then**
7:         **for** $g \in G$ **do**
8:             view_addition($V^g$, $V_{select}^g$, $S$, $N$);
9:         **end for**
10:        label all the added new views;
11:     **end if**
12: **end for**

---

---

**Algorithm 3** View Addition

---

1: **Input**: all views $V^g \in \{v_1^g, ..., v_n^g\}$ of scene $g$, selected view group $V_{select}^g$ of scene $g$, view selection score equation $S \in \{S_g, S_{mask}, S_{density}\}$, and task model $N$ ($= \emptyset$ if $S = S_g$).
2: $s\_v_{select} = -\inf$;
3: **for** $v \in V^g \setminus V_{select}^g$ **do**
4:     $s\_v = S(\{V_{select}^g, v\}, N)$;
5:     **if** $s\_v > s\_v_{select}$ **then**
6:         $v_{select} = v$;
7:         $s\_v_{select} = s\_v$;
8:     **end if**
9: **end for**
10: $V_{select}^g = \{V_{select}^g, v_{select}\}$.

---

**Training and testing.** For IVS, the training and testing processes are shown in Figure 6. The training process is described in the main manuscript (also in Algorithm 1 and 3), where the view selection, view labeling, and downstream model training are conducted independently. In the test, as shown at the bottom of Figure 6, the same view selection is conducted on the test set to select $K$ views. Then the selected views are fed into the well-trained downstream model for scene-level performance evaluation using the ground-truth constructed from all views.

For AVS, the training and testing processes are shown in Figure 7. In the training, the view selection, data labeling, and downstream task model training are conducted jointly, as shown in Algorithm 2 and 3. In the test, as shown at the bottom of Figure 7, no model training is required in the view selection process, and the model's prediction is directly used in the view selection score equation $S_{density}$ or $S_{mask}$. Note that for the multi-view crowd localization task, the $K$ views selected in the training stage are used directly for testing, as both MultiviewX and Widltrack are single-scene, multi-frame datasets with fixed camera views, so there is no need to conduct the view selection process again.

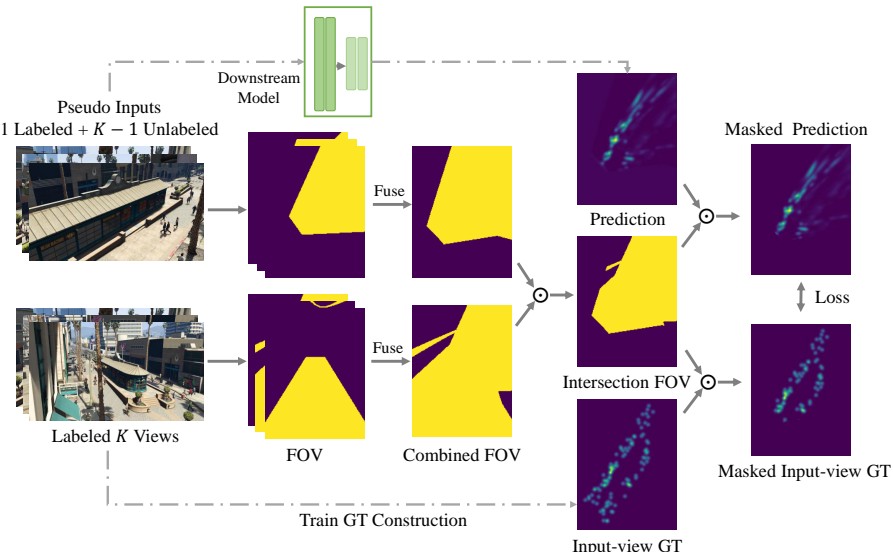

Figure 8: The pseudo label training during view selection. $\odot$ denotes the element-wise multiplication of matrices.

Figure 9: The pseudo label training in the final downstream model training after view selection. $\odot$ denotes the element-wise multiplication of matrices.

**Pseudo label generation and training.** To enhance the model's generalization ability, we utilize novel pseudo labels to train the downstream model better. During the view selection, the currently selected views $V_{select}^k$ and a random unselected view are combined as pseudo inputs to train the model, whose GT is ground-plane density maps of crowds covered by $V_{select}^k$. Specifically, as in Figure 8, for the pseudo inputs in the view selection, labeled $k$ views and 1 random view from the remaining unlabeled views will be together as the model's inputs (total $k + 1$ views) to obtain the predicted ground-plane density map, and the predicted density map is further masked by the combined FOV mask of the labeled $k$ views. The corresponding GT ground-plane density map is constructed from the labeled $k$ views, which is accurate for supervising the prediction masked with the combined FOV of the selected $k$ views. Thus, extra unlabeled views can be introduced in the downstream model training, enhancing its generalization to new views.

Besides, after the view selection, the selected views $V_{select}^K$ of the $F$ selected frames are used for downstream model training. In addition to that, we also add pseudo inputs in training, which is a mix of 1 selected view and $K-1$ unselected random views, whose pseudo-GT is the $K$ selected views' GT ground-plane density maps masked by the intersection of $H_v^K$ and the pseudo input views' combined FOV mask, and the prediction is also masked by the intersection FOV. Specifically, as shown in Figure 9, for the pseudo inputs after view selection, 1 selected view (random chosen from

$V_{select}^K$) and $K-1$ random views from the remaining unlabeled views are combined and regarded as the model's inputs to predict the ground-plane density map. The GT ground-plane density map is constructed from the labeled $K$ views. Since the pseudo inputs and GT are from different views, we can only supervise the common regions covered by the pseudo input views and the $K$-labeled views constructing the GT. Thus, we add a FOV intersection mask on both the prediction and GT density map in the loss calculation, where the intersection mask is the common region of the combined FOVs of the pseudo-input views and the $K$-labeled views. Note that both pseudo labels mentioned above, in the view selection or after view selection, are generated from the selected $F$ multi-frames and the labeled views.

By using pseudo labels, a large number of unlabeled views are incorporated into the model training, significantly enhancing the model's generalization capabilities. *Both IVS and AVS adopted pseudo labels in the model training.* For IVS, since it does not involve joint training with downstream tasks during the view selection process, the pseudo labels are only used in the final training of the downstream model after the view selection process. Additionally, for both IVS and AVS, the samples of the training set are doubled in the final model training stage after view selection. Specifically, compared with the original training method, which doesn't use the pseudo inputs, the pseudo inputs are used as additional samples, *i.e.* in the ratio of 1:1 for the $K$-labeled view inputs and the pseudo-label view inputs.

### A.2.1 MODEL AND TRAINING DETAILS.

For the multi-view counting model, we utilize the backbone model in CVCS (Zhang et al., 2021b) with a feature pyramid fusion network (FPN). In the backbone, the single-image feature extraction encoder utilizes the first 7 layers of VGG-Net (Simonyan & Zisserman, 2014). The outputs from the second, fourth, and seventh convolutional layers, along with a MaxPooling operation, are used in the multi-scale feature fusion of FPN. The single-view decoder resembles the multi-view decoder, with three additional convolutional layers of 512 channels in the middle layer. For the multi-view crowd localization model, we utilize the same MVDet (Hou et al., 2020) as implemented in MVSelect (Hou et al., 2024), which is trained with data augmentation.

The training losses consist of the single-view image density map loss and the ground-plane density map loss. For the downstream model training in view selection, the losses comprise the ground-plane density map loss and the single-image density map losses from the selected views $V_{select}^k$. In contrast, only the ground-plane density map loss is used for the multi-view crowd localization tasks. For final model training after view selection, the loss only consists of the ground-plane density map loss for pseudo inputs (due to no single-view GT), but both single-view and multi-view losses are used for labeled $K$-view inputs. The patch-based ground-plane density map is used for training, with a patch size of 160x180 and a patch time of 5 for each sample for multi-view crowd counting tasks. For multi-view crowd counting tasks, MSE is used for loss computation. For multi-view crowd localization tasks, similar to (Hou et al., 2024), only the ground-plane density map prediction loss is included, and the focal loss is used.

## B ADDITIONAL EXPERIMENTS

We conduct additional experiments to validate further the proposed view selection frameworks for multi-view crowd counting and localization tasks. For multi-view crowd counting, the default view number is $K = 5$, and the default multi-frame number is $F = 20$. For multi-view crowd localization, the view number $K$ is 3, and all the frames (360) from the training set are selected as the $F$ multi-frames.

### B.1 MULTI-VIEW CROWD COUNTING

**More performance evaluation.** We first use the crowd covering rate, denoted as 'CoverRate', as a metric to evaluate the view selection methods in Table 9, which indicates the percentage of the crowds covered by the selected views among all crowds in the scene. According to the table, our methods IVS/AVS have a higher CoverRate than all comparison methods. Moreover, even though IVS_$S_g$ has a close CoverRate to AVS_$S_{density}$, its scene-level counting performance is much worse than AVS_$S_{density}$. The reason is the view selection in AVS_$S_{density}$ is conducted

Table 9: Comparison of the multi-view counting results on the CVCS dataset.

| Method | GT_AllViews | | | GT_5Views | | | CoverRate↑ |
|---|---|---|---|---|---|---|---|
| | MAE ↓ | MSE ↓ | NAE ↓ | MAE ↓ | MSE ↓ | NAE ↓ | |
| MVMS (Random) (Zhang & Chan, 2019) | 36.65 | 43.03 | 0.271 | 22.55 | 26.97 | 0.191 | 0.885 |
| CVCS (Random) (Zhang et al., 2021b) | 39.18 | 44.92 | 0.289 | 25.74 | 30.42 | 0.213 | 0.885 |
| CountFormer (Random) (Mo et al., 2025) | 38.51 | 44.87 | 0.277 | 24.12 | 29.32 | 0.195 | 0.885 |
| Uniform | 21.76 | 25.75 | 0.163 | 20.35 | 23.64 | 0.162 | 0.945 |
| Uniform (Pseudo) | 15.69 | 19.92 | 0.115 | 11.26 | 14.09 | 0.089 | 0.945 |
| Random | 36.59 | 42.06 | 0.271 | 23.36 | 27.71 | 0.197 | 0.885 |
| Random (Pseudo) | 28.22 | 33.73 | 0.208 | 14.57 | 18.01 | 0.121 | 0.885 |
| Random (Oracle) | 15.37 | 20.91 | 0.115 | - | - | - | 0.885 |
| MVMS ($S_{sc}$) | 19.55 | 24.34 | 0.145 | 12.00 | 15.32 | 0.097 | 0.931 |
| IVS_$S_g$ (Baseline, Ours) | 14.98 | 18.93 | 0.111 | 10.59 | 13.38 | 0.082 | 0.959 |
| AVS_$S_{mask}$ (Ours) | 12.53 | 15.33 | 0.093 | **8.51** | **10.81** | **0.066** | 0.955 |
| AVS_$S_{density}$ (Ours) | **10.99** | **13.57** | **0.083** | 9.05 | 11.48 | 0.071 | **0.960** |

with the downstream model jointly, which takes the crowd density and location information and the view/scene geometries into account and *could select views more suitable for the downstream tasks, not only simply targeting at covering the crowd well*. Thus, AVS_$S_{density}$ achieves better performance and has stronger generalization ability.

In addition to using the predicted crowd count and the *scene-level* ground-truth count (all crowds in the scene) as metrics, i.e. 'GT_AllViews'. 'GT_5Views', which uses all the visible crowd in selected views as GT, is also used to measure the model's input-view-level counting performance. As shown in Table 9, our methods outperform the comparison methods, indicating that the proposed IVS and AVS frameworks not only can select views with more crowds in the scene, but also have better counting ability. Furthermore, even if AVS_$S_{mask}$ achieves higher input-view-level counting performance than AVS_$S_{density}$, the latter adopts both crowd location and density information in the view selection and for model training, so it achieves the best scene-level multi-view counting results.

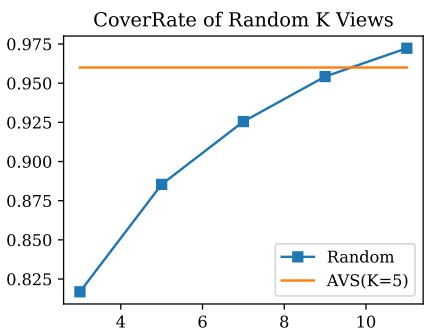

Figure 10: CoverRate of random $K$ views on CVCS dataset.

Moreover, we present the $S_g$ score, crowd coverage rate, and mean crowd coverage rate metrics of different view selection methods on CVCS, as shown in Figure 11. Our methods achieve higher $S_g$ scores, and AVS obtains the highest $CoverRate$ and $CoverRate_{mean}$, showing it selects better views covering both the scene and the crowd well. We also identify the specific number of views $(K)$ at which random selection's CoverRate approaches that of AVS shown in Figure 10. The CoverRate of random selection is the mean of 5 runs, indicating that AVS obtains views with a well CoverRate.

**More comparison methods.** We design several extra comparison methods in Table 9, denoted as 'CountFormer (Random)', 'Uniform', 'Uniform (Pseudo)', and 'MVMS ($S_{sc}$)'. 'CountFormer (Random)' uses CountFormer (Mo et al., 2025) as the multi-view counting model and the random view selection method for selecting views. 'Uniform' uses the same multi-view counting model as ours, but replaces the view selection method with the uniform view sampling from all views. 'Uniform (Pseudo)' means the use of pseudo labels during training. 'MVMS ($S_{sc}$)' uses MVMS (Zhang & Chan, 2019) as the multi-view counting model and only uses the scene coverage term $S_{sc}$ as the view selection score equation for selecting views. As the results show, we still outperform 'MVMS ($S_{sc}$)' and 'CountFormer (Random)', indicating that some strong downstream models with a relatively weak view selection strategy cannot solve scene-level tasks well. Moreover, as a view selection comparison method, 'Uniform' can slightly alleviate the occlusion problem in some scenes, but our method utilizes novel view selection methods together with a joint optimization framework for view selection and downstream model training, resulting in significantly improved scene-level performance.

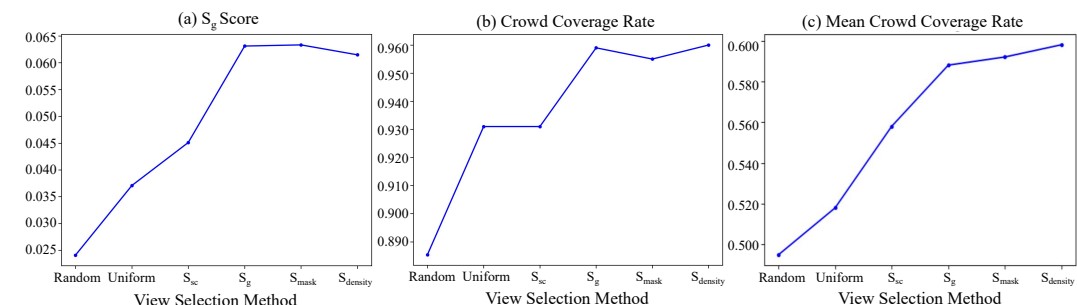

Figure 11: $S_g$ score, crowd coverage rate, and mean crowd coverage rate visualization.

Table 10: The ablation study on the multi-view counting models using AVS_$S_{density}$ on the CVCS dataset.

| Model | MAE ↓ | MSE ↓ | NAE ↓ |
|---|---|---|---|
| Backbone | 11.20 | 14.70 | 0.083 |
| Backbone+CVCS | 15.25 | 18.63 | 0.113 |
| Backbone+MVMS | **9.74** | **13.31** | **0.074** |
| Backbone+FPN | 10.99 | 13.57 | 0.083 |

**Ablation study on the different multi-view counting models.** The ablation studies on the multi-view counting models in the active view selection framework AVS_$S_{density}$ are shown in Table 10. The 'Backbone' model is the same backbone model of CVCS (Zhang et al., 2021b). '+MVMS','+CVCS', and '+FPN' mean adding the multi-view multi-scale selection (Zhang & Chan, 2019), camera view selection (Zhang et al., 2021b), and the feature pyramid fusion module (Lin et al., 2017) to the 'Backbone' model, respectively. From the table, using 'Backbone+MVMS' achieves the best multi-view counting results. 'Backbone+CVCS' achieves worse results than 'Backbone', perhaps due to the requirement of more labeled data to learn a good camera selection module, whereas our task setting provides limited labeled data. We adopt the 'Backbone+FPN' model as the multi-view counting model in our experiments for the balance of training efficiency and performance.

**Ablation study on the terms of $S_g$.** In addition to the term ablation study for $S_{density}$ of AVS framework in the manuscript, we conduct additional ablation experiments on the 3 terms in $S_g$ of the IVS framework in Table 11: using $S_{sc}$, $S_{sc} * S_{ad}$, $S_{sc} * S_{vd}$, or using all 3 terms (namely $S_g$). The results are similar to the experiments on $S_{density}$ and demonstrate that each term in $S_g$ contributes to the final performance by leveraging the scene and view geometries. Furthermore, only using $S_{sc}*S_{ad}$ with-

Table 11: The ablation study on the terms of the independent view selection score equation $S_g$ on the CVCS dataset.

| Term | MAE↓ | MSE↓ | NAE↓ |
|---|---|---|---|
| $S_{sc}$ | 18.80 | 23.82 | 0.139 |
| $S_{sc} * S_{ad}$ | 24.19 | 32.44 | 0.178 |
| $S_{sc} * S_{vd}$ | 18.32 | 22.68 | 0.135 |
| All ($S_g$) | **14.98** | **18.93** | **0.111** |

out the view diversity term $S_{vd}$ also yields worse results than $S_{sc}$, due to the similar view direction and locations of the selected views, which is ineffective for multi-view fusion.

**Ablation study on the testing view number.** We conduct extra experiments on the testing with different view numbers $K$ (3, 5, and 7) using the AVS_$S_{density}$ model trained with $K = 5$ views. As shown in Table 12, with the view number increasing, the model's performance improves relatively, as more regions are covered, demonstrating the good generalization ability of the proposed AVS framework to variable numbers of input views.

Table 12: The ablation study on the test of different selected view numbers $K$ using the AVS_$S_{density}$ model trained with $F = 20$ and $K = 5$ on the CVCS dataset.

| $K$ | MAE↓ | MSE↓ | NAE↓ |
|---|---|---|---|
| 3 | 15.06 | 18.88 | 0.112 |
| 5 | 10.99 | 13.57 | 0.083 |
| 7 | **10.53** | **13.09** | **0.080** |

**Ablation study on the frame number for view selection and computation time in testing on the CVCS dataset.** We conduct experiments on using different frame numbers in view selection when testing AVS_$S_{density}$ trained with labeled $F = 20$ frames and $K = 5$ views. Table 13 indicates that only using 20 frames at testing to conduct view selection can achieve almost similar results compared

Table 13: The ablation study on the frame number used for view selection when testing AVS_$S_{density}$ on the CVCS dataset. Note that the counting model still tests on all frames (100) for the result report.

| Frames | MAE↓ | MSE↓ | NAE↓ | Time (h)↓ |
|---|---|---|---|---|
| 5 | 11.43 | 14.20 | 0.085 | **2.0** |
| 20 | 10.97 | 13.60 | 0.082 | 7.1 |
| 50 | 10.91 | 13.54 | 0.082 | 15.6 |
| 100 | 10.99 | 13.57 | 0.083 | 30.7 |

Table 14: The ablation study on $\lambda$ of $S_{density}$ on the CVCS dataset.

| $\lambda$ | MAE | MSE | NAE |
|---|---|---|---|
| 0.05 | 12.88 | 16.24 | 0.096 |
| 0.1 | **10.99** | **13.57** | **0.083** |
| 0.5 | 15.07 | 18.18 | 0.112 |
| 1 | 12.74 | 15.87 | 0.093 |

Table 15: The ablation study on the threshold $\tau$ to conduct view addition on the CVCS dataset.

| $\tau$ | MAE | MSE | NAE |
|---|---|---|---|
| 15 | 11.47 | 13.99 | 0.086 |
| 20 | **10.99** | **13.57** | **0.083** |
| 30 | 12.79 | 15.66 | 0.096 |

with using 50 frames or 100 frames, with much less computation time, though. This demonstrates that the adopted frame initialization approach can effectively select representative frames for view selection and reduce testing computation time. Note that the counting model still tests on all frames (100) for the performance report.

**Ablation study on $\lambda$ in $S_{density}$.** We conduct experiments on the hype-parameter $\lambda$ in $S_{density}$ to validate the sensitivity of term $S_{vd}$. As shown in Table 14, compared to other settings, $\lambda = 0.1$ with intermediate sensitivity is more suitable for our score equation.

**Ablation study on the threshold $\tau$.** We conduct experiments on the downstream model performance threshold $\tau$ for conducting view addition. As shown in Table 15, $\tau = 20$ is more suitable for our AVS framework. When $\tau$ is too large, the counting model is under training, which may result in bad final performance; while when $\tau$ is too small, the counting model may be overfitted during each view addition step, also resulting in bad final performance. $\tau = 20$ can achieve a balance between the current counting model and final counting model performance.

**Ablation study on the number of labeled views for pseudo inputs.** We conduct experiments on the number of labeled views in pseudo inputs after view addition. With more labeled views in pseudo inputs, the randomness of the pseudo inputs is less, resulting in worse generalization ability, as shown in Table 16. Hence, we only retain one labeled view in pseudo inputs, achieving better performance.

**Ablation study on the ratio pseudo input.** We conduct experiments on the ratio between pseudo inputs and labeled inputs. Generally speaking, using more pseudo inputs can improve the model's robustness. However, due to the imperfect labels used, we cannot solely use the pseudo inputs for model training. As shown in Table 17, the ratio of 1:2 between labeled inputs and pseudo inputs achieves the best result, but we use the 1:1 ratio between labeled inputs and pseudo inputs for time tradeoff.

**Ablation study on frame selection method.** We conduct an additional random frame selection method, and our proposed frame selection method outperforms the random or uniform frame selection shown in Table 18. The reason is that our method selects frames across various lighting conditions using cosine similarity, thereby enhancing the sample diversity and the network's robustness. For uniform frame sampling, because of unordered frames on CVCS dataset, random frame sampling is equivalent to uniform frame sampling on CVCS dataset.

Table 18: The ablation studies on the random frame selection method on the CVCS dataset ($F = 20$).

| Frame Select | MAE | MSE | NAE |
|---|---|---|---|
| Random/Uniform | 12.28 | 15.46 | 0.092 |
| AVS (Ours) | **10.99** | **13.57** | **0.083** |

**Ablation study on DM-Count baseline and lighter single-image proxy.** To demonstrate the advantages of multi-view model architecture, we conduct the baseline with the IVS setting, which replaces the multi-view model with DM-Count and simply fuses the projected multi-view features by maximum. Compared to rows 1 and 3 shown in Table 19, the results indicate that the multi-view

Table 16: The ablation study on the fixed labeled view number in the pseudo inputs after view selection on the CVCS dataset.

| Fixed view num | MAE | MSE | NAE |
|---|---|---|---|
| 0 | 11.07 | 14.27 | 0.083 |
| 1 | **10.99** | **13.57** | **0.083** |
| 2 | 11.09 | 13.83 | 0.084 |
| 3 | 12.15 | 15.82 | 0.091 |
| 4 | 12.10 | 14.91 | 0.090 |

Table 17: The ablation study on the ratio between labeled inputs and pseudo inputs after view selection on the CVCS dataset.

| Ratio | MAE | MSE | NAE |
|---|---|---|---|
| Only Labeled | 12.65 | 15.35 | 0.096 |
| 1:1 | 10.99 | 13.57 | 0.083 |
| 1:2 | **10.85** | **13.56** | **0.080** |

Table 19: Comparison of the baseline of DM-Count and the lighter single-view proxy on the CVCS dataset.

| Row | Method | MAE ↓ | MSE ↓ | NAE ↓ |
|---|---|---|---|---|
| 1 | Multi-view Model (DMCount) | 20.38 | 26.43 | 0.155 |
| 2 | Lighter frame selector (MCNN) | 15.25 | 18.68 | 0.112 |
| 3 | IVS_$S_g$ (Ours) | 14.98 | 18.93 | 0.111 |
| 4 | AVS_$S_{density}$ (Ours) | **10.99** | **13.57** | **0.083** |

architecture outperforms a single-view model with a simple fusion strategy. Moreover, we also experimented using a lighter model, MCNN (Zhang et al., 2016), to pick frames with the IVS setting to verify the effect of the lighter single-image proxy for frame initialization. The results, compared with rows 1, 2, and 3, indicate that the multi-view model is vital, but the different single-image proxy frame selector has little effect on performance.

**Ablation study on comparison method with uniform view selection.** As shown in Table 20, the results with uniform view selection achieve better scene-level performance than those with random view selection, demonstrating the method's ability. However, compared to the comparison method, our methods obtain better results than uniform view selection, indicating that taking view/scene geometries and crowds' density level and location information into account enables selected views suitable for model prediction.

**Ablation study on comparison method with pseudo label.** We conduct pseudo-label for comparison methods, as shown in Table 21. The results indicate that the comparison method with a pseudo-label achieves better performance. However, compared with our methods, our methods' results are still better, demonstrating the advantage of our methods.

**Ablation study on comparison method with another random view selection.** Original methods with random view selection have the same random view groups. We also present another set of results from random view selection shown in Table 22. The results demonstrate that a well-designed view selection method is important for scene-level tasks with limited labels.

## B.2 MULTI-VIEW CROWD LOCALIZATION

**Comparison of training with different frame numbers.** We compare the proposed AVS_$S_{density}$ and MVSelect by training with different numbers of frames (36, 72, 180, and 360) on MultiviewX and Wildtrack in Table 23 and Table 24. The testing set is the same (40 frames). As the number of training frames increases, the performance improves, and our method outperforms MVSelect on various frame numbers, demonstrating the proposed method's efficiency over MVSelect.

**Comparison of training and testing with different view numbers.** We conduct experiments on the proposed method and MVSelect, training, or testing with different view numbers. For the results shown in Table 25 and Table 26, the view number used during both the training and testing processes is the same. As the number of views increases, the performance gradually becomes better, and it is nearly converged when $K \geq 3$. Furthermore, regardless of the number of views used, our method still outperforms MVSelect, indicating the advantage of the proposed method . For the results shown in Table 27 and Table 28, it uses the model trained with 3 views to test with different view numbers, $K = 2, 3, 4, 5$. Our method generally outperforms MVSelect on all view numbers (comparable on 5 views in Wildtrack), showing the proposed method's generalization ability to different input view

Table 20: The ablation study on the comparison method with uniform view selection.

| Method | MAE ↓ | MSE ↓ | NAE ↓ | CoverRate ↑ |
|---|---|---|---|---|
| MVMS (Random) | 36.65 | 43.03 | 0.271 | 0.885 |
| MVMS (Uniform) | 25.55 | 31.02 | 0.190 | 0.885 |
| CVCS (Random) | 39.18 | 44.92 | 0.289 | 0.885 |
| CVCS (Uniform) | 32.00 | 37.04 | 0.237 | 0.885 |
| Random | 36.59 | 42.06 | 0.271 | 0.885 |
| Uniform | 21.76 | 25.75 | 0.163 | 0.945 |
| IVS_$S_g$ (Baseline, Ours) | 14.98 | 18.93 | 0.111 | 0.959 |
| AVS_$S_{mask}$ (Ours) | 12.53 | 15.33 | 0.093 | 0.955 |
| AVS_$S_{density}$ (Ours) | **10.99** | **13.57** | **0.083** | **0.960** |

Table 21: The ablation study on the comparison method with pseudo-label.

| Method | MAE ↓ | MSE ↓ | NAE ↓ | CoverRate ↑ |
|---|---|---|---|---|
| MVMS (Random) | 36.65 | 43.03 | 0.271 | 0.885 |
| MVMS (Random, Pseudo) | 29.94 | 35.85 | 0.221 | 0.885 |
| CVCS (Random) | 39.18 | 44.92 | 0.289 | 0.885 |
| CVCS (Random, Pseudo) | 30.41 | 36.43 | 0.223 | 0.885 |
| Uniform | 21.76 | 25.75 | 0.163 | 0.945 |
| Uniform (Pseudo) | 15.69 | 19.92 | 0.115 | 0.945 |
| Random | 36.59 | 42.06 | 0.271 | 0.885 |
| Random (Pseudo) | 28.22 | 33.73 | 0.208 | 0.885 |
| IVS_$S_g$ (Baseline, Ours) | 14.98 | 18.93 | 0.111 | 0.959 |
| AVS_$S_{mask}$ (Ours) | 12.53 | 15.33 | 0.093 | 0.955 |
| AVS_$S_{density}$ (Ours) | **10.99** | **13.57** | **0.083** | **0.960** |

numbers. Note that MVSelect utilizes labels from all views for training and is challenging to apply to new scenes due to the reinforcement learning framework.

**Comparison of frame selection method.** Compared with the uniform and random frame selection methods in Table 29, our frame selection approach is still better. As mentioned above, our method selects frames with various lighting conditions using cosine similarity, which can enhance the diversity of the sample and the robustness of the network.

**Comparison of view selection method.** Compared with uniform and random view selection methods in Table 30, our method generally achieves the best result because of considering the view/scene geometries, crowd density-level information, and location information.

**Comparison of the costs.** As shown in Table 31, Due to using pseudo labels and model predictions for view selection, our training time is higher than MVSelect and Random, but comparable to others. Yet, our test speed is similar to baselines and faster than MVSelect since no extra reinforcement learning network is needed.

**Comparison of threshold $\tau$.** As a hyperparameter, the threshold $\tau$ controls when to select the next view. For the multi-view crowd localization task on Wildtrack and MultiviewX, AVS framework will select the next view when the MODA during training exceeds the threshold $\tau$. AVS framework jointly optimizes view selection and downstream task models. Hence, a well-trained model can produce a more accurate density map, thereby reducing prediction errors during view selection. Consequently, we need an appropriate model to select views by controlling the threshold $\tau$. As shown in Table 32, threshold $\tau = 40$ is more suitable on Wildtrack and MultiviewX, achieving the better scene-level performance,

**Experiment on CVCS dataset.** We conduct localization experiment on CVCS dataset shown in Table 33 with a similar setting of counting including model, $K = 5$, and $F = 20$. And MODP with distance threshold t = 0.5m. The results with our view selection method are better than those with other view selections (MODA and F1 score are the main metrics for overall performance), further demonstrating our method's advantages.

Table 22: The ablation study on the comparison method with another random view selection.

| Method | MAE ↓ | MSE ↓ | NAE ↓ | CoverRate ↑ |
|---|---|---|---|---|
| MVMS (Random) | 36.65 | 43.03 | 0.271 | 0.885 |
| MVMS (Random) | 37.22 | 44.74 | 0.276 | 0.872 |
| CVCS (Random) | 39.18 | 44.92 | 0.289 | 0.885 |
| CVCS (Random) | 24.27 | 30.59 | 0.179 | 0.901 |
| Uniform | 21.76 | 25.75 | 0.163 | 0.945 |
| Random | 36.59 | 42.06 | 0.271 | 0.885 |
| IVS_$S_g$ (Baseline, Ours) | 14.98 | 18.93 | 0.111 | 0.959 |
| AVS_$S_{mask}$ (Ours) | 12.53 | 15.33 | 0.093 | 0.955 |
| AVS_$S_{density}$ (Ours) | **10.99** | **13.57** | **0.083** | **0.960** |

Table 23: The ablation study on the training frame number $F$ on the MultiviewX dataset.

| Method | AVS_$S_{density}$ | | | | | MVSelect | | | | |
|---|---|---|---|---|---|---|---|---|---|---|
| Frame | MA. | MP. | P | R | F1 | MA. | MP. | P | R | F1 |
| 36 | 73.6 | 76.4 | 95.8 | 77.0 | 85.4 | 61.5 | 73.4 | 96.9 | 63.5 | 76.7 |
| 72 | 81.2 | 71.6 | 95.9 | 84.8 | 90.0 | 69.8 | 51.2 | 96.6 | 72.4 | 82.8 |
| 180 | 86.0 | 80.8 | 96.7 | 89.1 | 92.7 | 77.6 | 60.0 | 97.1 | 80.0 | 87.7 |
| 360 | **89.2** | **82.1** | **98.0** | **91.0** | **94.4** | **88.1** | **89.8** | **98.2** | **89.7** | **93.8** |

## C  EXTRA VISUALIZATIONS

**Multi-view crowd counting.** As shown in Figure 12, compared with other methods, our method can cover more crowds in the scene (red dots in 'CamGeometry' indicate crowds not covered by the selected views), and achieve better scene-level counting performance. IVS cannot cover all crowds, because crowds covered by the camera FOV area cannot be directly regarded as visible crowds due to the occlusion in the scene. Moreover, the view score $S_g$ of the 8 methods in Figure 12 are: 0.035, 0.021, 0.052, 0.030, 0.008, 0.072, 0.063, and 0.063, showing our methods achieve higher $S_g$ scores in this example. The crowds on the bottom right of View 2 of IVS_$S_g$ are occluded by the building even though View 2's FOV region covers them according to $c_2$ in CamGeometry line. AVS_$S_{density}$ achieves the best result by considering both crowd density-level information and location information, and the view/scene geometries in the view selection.

**Multi-view crowd localization.** We visualize the results from comparison methods and proposed methods on MultiviewX and Wildtrack in Figure 13 and 14. Because both are smaller datasets compared with CVCS, a few views can easily cover the crowd on the ground. Our method achieves better scene-level counting performance than the comparisons, as shown in the red box regions, where our methods have fewer missing points. Our AVS framework jointly optimizes view selection and the multi-view localization task, and adopts novel pseudo labels during model training to achieve better localization performance.

Table 24: The ablation study on the training frame number $F$ on the Wildtrack dataset.

| Method | AVS_$S_{density}$ | | | | | MVSelect | | | | |
|---|---|---|---|---|---|---|---|---|---|---|
| Frame | MA. | MP. | P | R | F1 | MA. | MP. | P | R | F1 |
| 36 | 72.5 | 66.4 | 91.5 | 79.9 | 85.3 | 65.7 | 68.7 | **96.2** | 68.4 | 80.0 |
| 72 | 79.2 | 62.9 | 95.3 | 83.3 | 88.9 | 73.6 | 71.0 | 94.9 | 77.8 | 85.5 |
| 180 | 83.7 | 72.5 | **96.2** | 87.2 | 91.5 | 78.7 | 61.5 | 93.5 | 84.6 | 88.8 |
| 360 | **89.6** | **76.7** | 96.1 | **93.4** | **94.7** | **88.6** | **79.9** | 93.3 | **94.2** | 93.7 |

Table 25: The ablation study on the selected view number $K$ on the MultiviewX dataset, training and testing using the same $K$ views.

| Method | AVS_$S_{density}$ | | | | | MVSelect | | | | |
|---|---|---|---|---|---|---|---|---|---|---|
| View | MA. | MP. | P | R | F1 | MA. | MP. | P | R | F1 |
| 2 | 81.7 | 80.4 | 97.1 | 84.3 | 90.2 | 73.9 | 71.4 | 97.2 | 76.1 | 85.4 |
| 3 | 89.2 | **82.1** | 98.0 | 91.0 | 94.4 | 88.1 | **89.8** | 98.2 | 89.7 | 93.8 |
| 4 | 92.0 | 78.7 | 98.2 | 93.7 | 95.9 | 87.1 | 78.5 | 97.5 | 89.3 | 93.2 |
| 5 | **93.4** | 79.2 | **98.3** | **95.1** | **96.6** | **90.7** | 77.8 | **98.3** | **92.3** | **95.2** |

Table 26: The ablation study on the selected view number $K$ on the Wildtrack dataset, training and testing using the same $K$ views.

| Method | AVS_$S_{density}$ | | | | | MVSelect | | | | |
|---|---|---|---|---|---|---|---|---|---|---|
| View | MA. | MP. | P | R | F1 | MA. | MP. | P | R | F1 |
| 2 | 82.0 | 74.4 | **96.5** | 85.1 | 90.4 | 74.8 | 76.5 | 93.6 | 80.3 | 86.4 |
| 3 | **89.6** | 76.7 | 96.1 | 93.4 | **94.7** | **88.6** | **79.9** | 93.3 | **94.2** | **93.7** |
| 4 | 89.0 | 77.7 | 94.3 | 94.8 | 94.5 | 85.6 | 69.3 | **93.8** | 91.6 | 92.7 |
| 5 | 89.0 | **78.0** | 93.5 | **95.6** | 94.5 | 87.1 | 60.8 | 93.0 | 94.2 | 93.6 |

Table 27: The ablation study on the selected view number in testing on MultiviewX dataset, training with 3 views and testing with $K$ (2, 3, 4, and 5) views.

| Method | AVS_$S_{density}$ | | | | | MVSelect | | | | |
|---|---|---|---|---|---|---|---|---|---|---|
| View | MA. | MP. | P | R | F1 | MA. | MP. | P | R | F1 |
| 2 | 82.8 | 80.4 | 97.0 | 85.5 | 90.9 | 77.1 | 83.4 | 96.5 | 80.0 | 87.5 |
| 3 | 89.2 | 82.1 | 98.0 | 91.0 | 94.4 | 88.1 | **89.8** | 98.2 | 89.7 | 93.8 |
| 4 | 92.7 | **82.6** | 98.5 | 94.2 | 96.3 | 91.2 | 87.8 | 98.3 | 92.9 | 95.5 |
| 5 | **93.4** | 82.4 | **98.6** | **94.7** | **96.6** | **93.1** | 88.9 | **98.6** | **94.5** | **96.5** |

Table 28: The ablation study on the selected view number in testing on the Wildtrack dataset, training with 3 views and testing with $K$ (2, 3, 4, and 5) views.

| Method | AVS_$S_{density}$ | | | | | MVSelect | | | | |
|---|---|---|---|---|---|---|---|---|---|---|
| View | MA. | MP. | P | R | F1 | MA. | MP. | P | R | F1 |
| 2 | 84.9 | 76.1 | 95.8 | 88.8 | 92.1 | 76.0 | 75.9 | **93.8** | 81.3 | 87.1 |
| 3 | 89.6 | 76.7 | **96.1** | 93.4 | 94.7 | 88.6 | 79.9 | 93.3 | 94.2 | 93.7 |
| 4 | 89.7 | 76.7 | 93.7 | 96.2 | 94.9 | 89.6 | 80.3 | 93.6 | 96.2 | 94.9 |
| 5 | **90.0** | **78.0** | 93.7 | **96.5** | 95.1 | **90.0** | **81.2** | 93.5 | **96.7** | 95.1 |

Table 29: The ablation study on the same training frame number $F = 72$ with different frame selection methods. IDdiff consists of the selected frame ID mean and standard deviation.

| Dataset | Wildtrack | | | MultiviewX | | |
|---|---|---|---|---|---|---|
| Frame Select | MA. | F1 | IDdiff | MA. | F1 | IDdiff |
| Random | 77.5 | 87.8 | 24.7± 20.7 | 80.7 | 89.6 | 4.9±4.1 |
| Uniform | 77.0 | 87.5 | 25.0± 0.0 | 80.8 | 89.6 | 5.0±0.0 |
| AVS(Ours) | **79.2** | **88.9** | 18.0 ±52.3 | **81.2** | **90.0** | 5.0±6.0 |

Table 30: The ablation study on the different view selection methods.

| Dataset | MultiviewX | | | | | Wildtrack | | | | |
|---|---|---|---|---|---|---|---|---|---|---|
| View Select | MA. | MP. | P | R | F1 | MA. | MP. | P | R | F1 |
| Random | 85.3 | 80.8 | 97.3 | 87.7 | 92.2 | 80.6 | 75.8 | 93.0 | 87.1 | 89.8 |
| Uniform | 82.6 | **87.3** | 96.4 | 85.8 | 90.8 | 84.6 | **79.7** | 95.1 | 89.2 | 92.0 |
| AVS_$S_{density}$ (Ours) | **89.2** | 82.1 | **98.0** | **91.0** | **94.4** | **89.6** | 76.7 | **96.1** | **93.4** | **94.7** |

Table 31: Cost comparison on MultiviewX.

| Method | Memory(GB) | FLOPs(G) | Train(s) | Test(s) |
|---|---|---|---|---|
| MVSelect | 16.879 | 532.200 | 1200 | 10 |
| Random | 17.594 | 530.703 | 1700 | 7 |
| Random (Pseudo) | 17.594 | 530.703 | 7600 | 7 |
| IVS_$S_g$ (Ours) | 17.594 | 530.703 | 8006 | 7 |
| AVS_$S_{mask}$ (Ours) | 17.594 | 530.703 | 9172 | 7 |
| AVS_$S_{density}$ (Ours) | 17.594 | 530.703 | 9194 | 7 |

Table 32: The ablation study on the threshold $\tau$ to conduct view addition on the MultiviewX and Wildtrack.

| Dataset | MultiviewX | | | | | Wildtrack | | | | |
|---|---|---|---|---|---|---|---|---|---|---|
| $\tau$ | MA. | MP. | P | R | F1 | MA. | MP. | P | R | F1 |
| 30 | 87.5 | 81.3 | 97.3 | 90.0 | 93.5 | 85.3 | 76.3 | 96.0 | 89.0 | 92.4 |
| 40 | **89.2** | **82.1** | 98.0 | **91.0** | **94.4** | **89.6** | **76.7** | 96.1 | **93.4** | **94.7** |
| 50 | 87.4 | 78.3 | **98.1** | 89.1 | 93.4 | 87.4 | 76.1 | **96.4** | 90.8 | 93.5 |
| 60 | 87.0 | 82.1 | 97.9 | 88.9 | 93.2 | 86.9 | 73.6 | 96.0 | 90.7 | 93.2 |

Table 33: The multi-view crowd localization results on the CVCS dataset. We are the best according to the main metrics, MODA and F1 score.

| Method | MODA↑ | MODP↑ | Precision↑ | Recall↑ | F1_score↑ |
|---|---|---|---|---|---|
| Random | 15.4 | 58.7 | 72.4 | 24.9 | 37.1 |
| Uniform | 24.2 | 61.0 | 80.9 | 31.6 | 45.5 |
| Uniform (Pseudo) | 27.6 | **62.7** | **83.6** | 34.4 | 48.7 |
| IVS | 27.8 | 62.1 | 81.6 | 36.0 | 50.0 |
| AVS_$S_{density}$ | **29.4** | 62.4 | 82.8 | **37.1** | **51.3** |

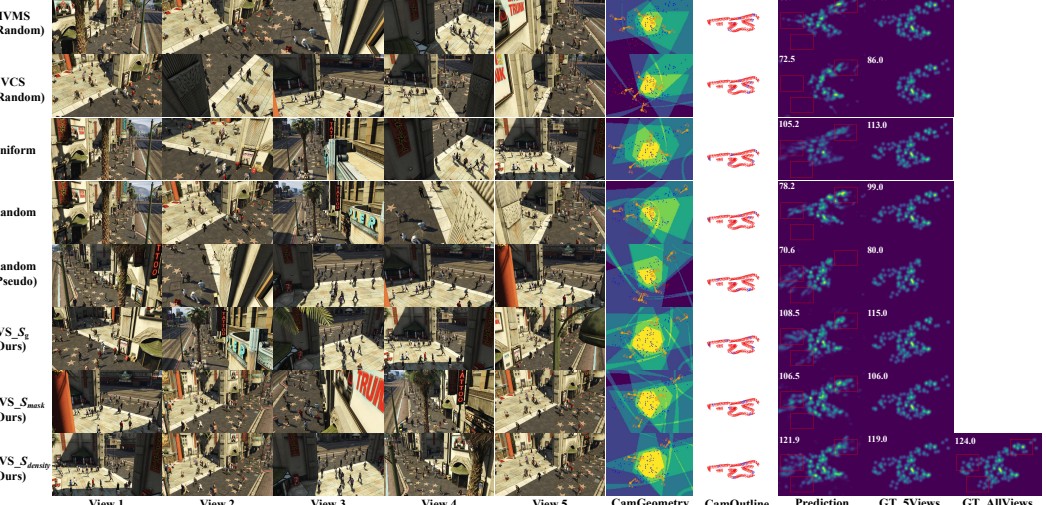

Figure 12: The view selection and multi-view counting results on the CVCS dataset with the view's location and direction. $c_j$ and $o_j$ in CamGeometry represent the $j$-th view location and direction on the ground respectively. Red dots are uncovered crowds, not visible by the selected views. Blue camera view indicates the selected view in column CamOutline.

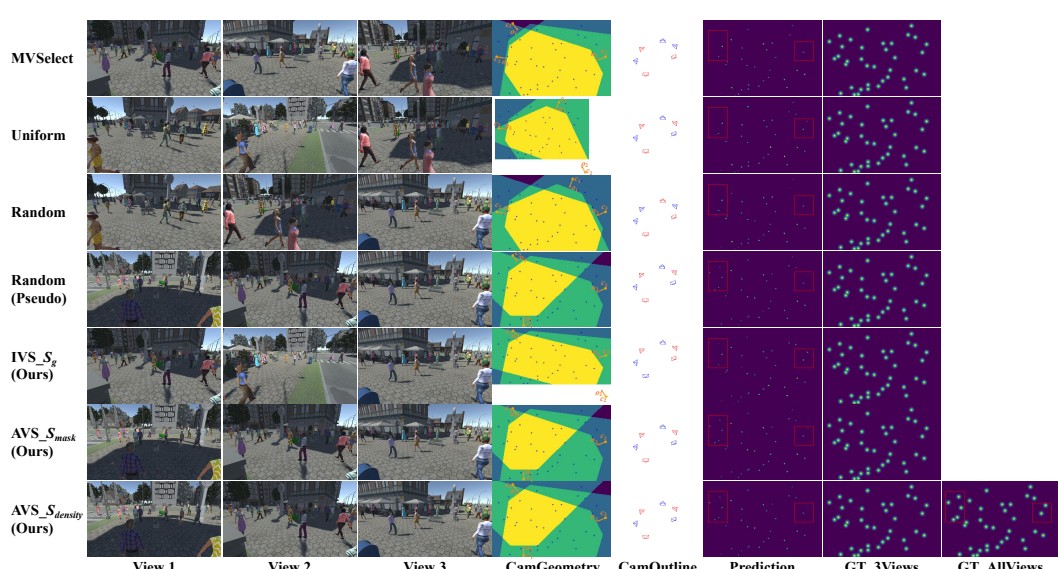

Figure 13: The view selection and multi-view localization results on the MultiviewX dataset. Blue camera view indicates the selected view in column CamOutline. Zoom in for better views.

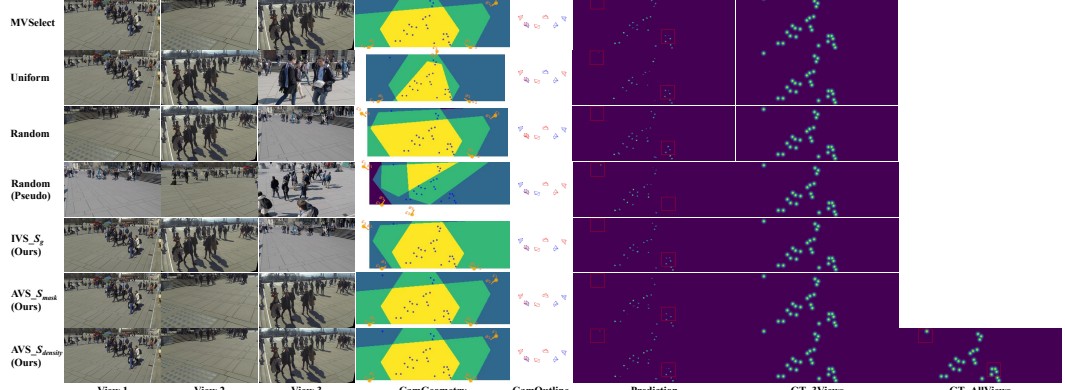

Figure 14: The view selection and multi-view localization result on the Wildtrack dataset. Blue camera view indicates the selected view in column CamOutline. Zoom in for better views.

