# OpenReview forum: "Active View Selection for Scene-level Multi-view Crowd Counting and Localization with Limited Labels"
_ICLR.cc/2026/Conference — Submitted to ICLR 2026_

### Official Review · Reviewer_YB4N · 2025-10-30

**Soundness:** 3
**Presentation:** 3
**Contribution:** 2
**Rating:** 4
**Confidence:** 4

**Summary:**

This paper achieves scene-level multi-view crowd counting and localization with limited labeled views. It introduces an Independent View Selection baseline (IVS) that selects views based on view and scene geometries, and an Active View Selection framework (AVS) that jointly optimizes view selection and downstream task models. Experiments on three benchmarks, including CVCS, Wildtrack, and MultiviewX, demonstrate that AVS outperforms existing methods with limited labels.

**Strengths:**

**Innovative and effective approach**: AVS first introduces the view selection into multi-view crowd counting and localization with reduced labels, jointly optimizing view selection and downstream tasks, improving scene-level coverage and model accuracy.

**Good performance**: AVS achieves SOTA performance beyond existing methods with randomly selected views in counting, and even comparable with full labels in localization, exhibiting a great generalization ability.

**Weaknesses:**

**Heuristic Design and Sensitivity**: The design of the view-scoring function S seems heuristic and implicitly assumes relatively uniform or overlapping camera placement., However, the sensitivity to camera geometries has not been thoroughly examined.

**Insufficient experiments and unfair experimental setups**: In Table 1, IVS and AVS achieve similar CoverRate values, yet exhibit a significant performance gap. The paper does not provide sufficient analysis to explain this discrepancy.

The direct comparison between AVS and SOTA methods (with all views) under 5 views is unfair. The key to demonstrating AVS's utility is to show the "view-to-performance" curve, specifically reporting the K required to achieve SOTA-level accuracy, which truly reflects its annotation efficiency.

**Unclear mechanism**: The random selection of unlabeled views for pseudo-label generation does not account for their information complementarity or redundancy with labeled views.

**Questions:**

It is recommended to visualize the full camera layout and explicitly compare selected versus unselected views to better illustrate the behavior and selection characteristics of the AVS strategy.

The five views in the first few random selection strategies shown in Table 1 are different, but why do they all have a CoverRate of 0.885 in the end?

In Table 1, CoverRate is used to measure the quality of random versus strategy-based view selection. However, as the number of randomly chosen views increases, CoverRate naturally improves and may approach that of guided selection. The authors should identify the specific number of views (K) at which random selection's CoverRate approaches that of AVS.

The method still seems to require labels to calculate the score for the view selection, otherwise how to get the FOV.

---

> ### Author Response · Authors · 2025-11-29
> **Response to Reviewer YB4N (Part 1/2)**
>
> Thank you for your valuable suggestions. Each question is answered as follows. We hope the response has addressed your concerns. We're glad to have further communication with you. The revision in the paper is marked in blue.
>
> ### **Q1: Heuristic Design, sensitivity to camera geometries, and the visualisation of the full camera layout.**
>
> Thank you for your question.
>
> By leveraging view/scene geometries, we propose IVS, which takes scene coverage, average distance, and view diversity into account. Scene coverage ensures the selected views cover the entire scene as much as possible. The average distance prevents the target scale in the camera views from becoming too small, thereby mitigating the effect on the model's prediction. As in previous work, view diversity can prevent all selected views from converging, thereby enhancing the effect of multiple views and reducing ambiguity.
>
> Based on IVS, we propose AVS, which jointly optimizes view selection and downstream task models. During view selection, AVS further considers crowd localization and crowd density map information. Using the mask generated from the crowd density map can more accurately indicate the regions where crowds appear in the scene, enabling the selected views to cover as many people as possible. Moreover, the crowd density map can identify crowded areas, and areas with higher density should have higher weights in the view selection process. Extensive experiments on the two tasks reveal the advantages of the proposed method.
>
> To assess sensitivity to camera geometry, we conducted experiments across various view numbers in the cross-scene dataset CVCS (see Table 5 and 12), demonstrating the insensitivity of the proposed view selection methods to different camera view distributions.
>
> Additionally, we present the full camera layout in Figure 4, 12, 13, 14 to better illustrate the characteristics of the selected views. As shown in the columns CamGeometry and CamOutline in Figure 4, 12, 13, 14, the selected views of AVS generally have full coverage of the whole scene, a scattered camera distribution, greater crowd coverage, and greater multi-view overlap. And the results shown in Table 2 demonstrate its advantages.
>
> ### **Q2: Insufficient experiments and unfair experimental setups: In Table 1, IVS and AVS achieve similar CoverRate values, yet exhibit a significant performance gap. The paper does not provide sufficient analysis to explain this discrepancy.**
>
> This is a misunderstanding of our task and metrics. CoverRate is used as a metric to determine whether the selected view covers as many people in the scene as possible. However, for current scene-level tasks with limited labels, we need to consider both selecting views and training the model. Consequently, we proposed IVS and AVS. IVS takes view and scene geometries into account. Based on IVS, AVS considers downstream model predictions in view selection and jointly optimizes view selection and downstream tasks, achieving better scene-level performance even when its CoverRate is similar. Experiments on the two tasks (see Table 2 and 7) reveal the advantages of the proposed AVS method.
>
> ### **Q3: The direct comparison between AVS and SOTA methods (with all views) under 5 views is unfair. The key to demonstrating AVS's utility is to show the "view-to-performance" curve, specifically reporting the K required to achieve SOTA-level accuracy, which truly reflects its annotation efficiency.**
>
> We argue that our goal is to achieve better scene-level task performance with limited labels, which requires selecting better views and training better models. Hence, we proposed AVS framework to jointly optimize the view selection and model training to achieve better scene-level model performance under limited labels. Moreover, we also conduct experiments on the number of views. As shown in Table 5, 12, 25-28, as the number of views increases, the model’s performance generally improves, indicating that the proposed AVS framework generalizes well across variable numbers of input views.

---

> ### Author Response · Authors · 2025-11-29
> **Response to Reviewer YB4N (Part 2/2)**
>
> ### **Q4: Unclear mechanism: The random selection of unlabeled views for pseudo-label generation does not account for their information complementarity or redundancy with labeled views.**
>
> By leveraging pseudo-labels, we add pseudo-inputs to the training set: either a mix of 1 selected view and $K-1$ unselected random views after view selection, or a mix of 1 random unlabeled view and $k$ selected views during view selection. During training, instead of using only labeled data, all views, including labeled and unlabeled, will be used, expanding the capacity of the training data and reducing the need for labels. We also conduct experiments on the view composition and moment to demonstrate the efficiency, as shown in Table 6, 8, 16, 17. See Lines 902-935 for more details about pseudo-label generation.
>
> ### **Q5: Random selection strategies have same CoverRate.**
>
> Thank you for pointing this out. These methods with random view selection have the same random view groups. We also present another set of results from random view selection. The results shown below (or see Table 22) demonstrate that a well-designed view selection method is important for scene-level tasks with limited labels.
>
> | Method | MAE $\downarrow$ | MSE $\downarrow$ | NAE $\downarrow$ | CoverRate $\uparrow$ |
> |--------|------------------|------------------|------------------|---------------------|
> | MVMS (Random) | 37.22 | 44.74 | 0.276 | 0.872 |
> | CVCS (Random) | 24.27 | 30.59 | 0.179 | 0.901 |
> | Random | 36.59 | 42.06 | 0.271 | 0.885 |
> | IVS\_$S_g$ (Baseline, Ours) | 14.98 | 18.93 | 0.111 | 0.959 |
> | AVS\_$S_{mask}$ (Ours) | 12.53 | 15.33 | 0.093 | 0.955 |
> | AVS\_$S_{density}$ (Ours) | **10.99** | **13.57** | **0.083** | **0.960** |
>
> ### **Q6: CoverRate influence.**
>
> CoverRate is just a metric that partially reflects the quality of selected views. But for scene-level tasks with limited labels, we need to consider both selecting views and training the model. Hence, we propose the AVS framework that jointly optimizes the view selection and downstream task models to achieve better scene-level performance. We also conduct experiments about the specific number of views with random view selection to approach AVS. As shown below (or see Figure 10), the CoverRate for random view selection is the mean of 5 runs, and the results indicate that AVS obtains the views with a high CoverRate with the same view number.
>
> | Method | $K$ | CoverRate $\uparrow$ |
> |--------|-----|----------------------|
> | Random | 3 | 0.817 |
> | Random | 5 | 0.885 |
> | Random | 7 | 0.925 |
> | Random | 9 | 0.954 |
> | Random | 11 | 0.972 |
> | AVS | 5 | 0.960 |
>
> ### **Q7: Relationship between FOV and labels.**
>
> Thank you for your question. Because the FOV is calculated based on the scene cover rate rather than people, we only need the camera pose to complete the projection.

---

### Official Review · Reviewer_LBiA · 2025-10-31

**Soundness:** 2
**Presentation:** 2
**Contribution:** 2
**Rating:** 2
**Confidence:** 4

**Summary:**

This paper tackles scene-level multi-view crowd counting and localization with limited labels. Given a budget of F annotated frames per camera view and K camera views per scene, the key idea is to identify most informative camera views for scene-level prediction. Specifically, the authors introduce an activate view selection framework, which jointly optimizes view selection and downstream model training in an iterative manner. Evaluations on three multi-view counting datasets show that the proposed method achieves state-of-the-art results.

**Strengths:**

* The paper investigates camera view selection in multi-view crowd counting, which could be a promising way to reduce annotation costs.
* An active view selection framework is proposed to jointly optimizes camera view selection and downstream models, achieving promising results.
* The proposed method could be applied to new scenes by adaptively selecting optimal camera views.

**Weaknesses:**

* Unconvincing results in Table 1. First, it is unfair to compare with previous methods under random view setting. Random view selection inevitably discards potentially informative views, leading to degraded results. For a fair comparison, the authors should compare with existing methods under Uniform setting, where camera views are uniformly sampled. Second, the rationale of computing scene-level counting error remains questionable. Intuitively, using a few input views inherently restricts observable content. Evaluating performance on visible input views is more reasonable. Third, the authors should also conduct experiments on real-world multi-view counting datasets, e.g., CityStreet, DukeMTMC, and PETS2009.
* The effectiveness of the proposed method requires further justification. Table 7 shows that the proposed view selection strategy only achieves marginal improvement in CoverRate over naïve Uniform baseline. In addition, Tables 17 and 18 indicate that MVSelect could achieve similar performance when provided with more training frames. These results raise concerns regarding the necessity and superiority of the proposed method.
* High computational overhead. The proposed method requires an iterative process to select K camera views during training, followed by a separate training phase for downstream counting model using finalized camera views.
As shown in Table 25, this leads to a 6–7x increase in training time compared to MVSelect.
* It would be better to include cross-domain evaluations to support the generalization capability of the proposed method.

**Questions:**

* Did compared method adopt pseudo labels during training? It appears that pseudo labeling is important to improve performance.

---

> ### Author Response · Authors · 2025-11-29
> **Response to Reviewer LBiA (Part 1/2)**
>
> Thank you for your valuable suggestions. Each question is answered as follows. We hope the response has addressed your concerns. We're glad to have further communication with you. The revision in the paper is marked in blue.
>
> ### **Q1: Compare with existing methods under uniform view selection; conduct experiments on real-world multi-view counting datasets.**
>
> Thank you for pointing this out and your suggestion. The results are as follows.
>
> | Row | Method | MAE $\downarrow$ | MSE $\downarrow$ | NAE $\downarrow$ |
> |-----|--------|------------------|------------------|------------------|
> | 1 | CVCS (Random Selection) | 39.18 | 44.92 | 0.289 |
> | 2 | CVCS (Uniform Selection) | 32.00 | 37.04 | 0.237 |
> | 3 | MVMS (Random Selection) | 36.65 | 43.03 | 0.271 |
> | 4 | MVMS (Uniform Selection) | 25.55 | 31.02 | 0.190 |
> | 5 | IVS\_$S_g$ (Ours) | 14.98 | 18.93 | 0.111 |
> | 6 | AVS\_$S_{mask}$ (Ours) | 12.53 | 15.33 | 0.093 |
> | 7 | AVS\_$S_{density}$ (Ours) | **10.99** | **13.57** | **0.083** |
>
> For uniform view selection, comparing rows 1 and 2 (or rows 3 and 4, or see Table 20), the results with uniform view selection achieve better scene-level performance than those with random view selection, demonstrating the method's ability. However, compared to rows 2 and 4 and our methods, our methods obtain better results than uniform view selection, indicating that taking view/scene geometries and crowds' density level and location information into account enables selected views suitable for model prediction.
>
> For real-world multi-view counting datasets, we also conduct experiments on CityStreet with $K=2$ and $F=60$ for training, and $K=2$ for testing. As shown below, compared to rows 1 and 2 (or see Table 4), results with uniform view selection are better than random view selection. Moreover, our methods achieve the best results, indicating that equipping a well-designed view selection method is essential for scene-level tasks with limited labels.
> | Row | Method | MAE $\downarrow$ | MSE $\downarrow$ | NAE $\downarrow$ |
> |-----|--------|------------------|------------------|------------------|
> | 1 | Random  | 13.47 | 16.60 | 0.170 |
> | 2 | Uniform  | 11.82 | 15.00 | 0.130 |
> | 3 | IVS  | 11.28 | 14.36 | 0.128 |
> | 4 | AVS\_$S_{density}$  | **9.80** | **11.93** | **0.118** |
>
>
> ### **Q2: The rationale of scene-level counting.**
>
> This is a misunderstanding for our task. Generally speaking, previous methods always focus on view-level performance, leading to greater attention to model design. However, scene-level tasks with limited labels not only reflect the model's performance but also measure the quality of selected views. In this situation, we need to select a limited set of views that cover the scene objects as much as possible while achieving good performance with limited labels, which requires selecting appropriate views and efficiently training downstream task models that fit them. Consequently, we propose the active view selection method (AVS) jointly optimizes view selection and downstream tasks, achieving better scene-level performance with limited labels. See Lines 34-109 for more descriptions.
>
> ### **Q3: The effectiveness of the proposed method requires further justification.**
>
> Thank you very much for pointing that out.
>
> Commonly, CoverRate is used as a metric to determine whether the selected view covers all people in the scene as much as possible. However, for current scene-level tasks with limited labels, we need to consider both selecting views and training the model. Consequently, we proposed IVS and AVS. IVS considers view and scene geometries and optimizes the view selection and downstream model separately. Based on IVS, AVS considers downstream model predictions in view selection and jointly optimizes view selection and downstream tasks, achieving better scene-level performance even when its CoverRate is similar to uniform baseline (see Table 2).
>
> As for MVSelect, it can't be directly applied to new scenes with different camera views due to its reinforcement-learning architecture. MVSelect needs to ensure a fixed action space for the next perspective selection, leading to the necessity of a fixed number of views and all the labels for training. However, in cross-scene (CVCS) datasets, the number of views per scene varies, which limits MVSelect's applicability. Consequently, we proposed IVS and AVS, which can be applied in cross-scene datasets with limited labels. Additionally, by jointly optimizing view selection and model training, AVS can achieve better performance on cross-scene datasets with limited labels.

---

> ### Author Response · Authors · 2025-11-29
> **Response to Reviewer LBiA (Part 2/2)**
>
> ### **Q4: High computational overhead.**
>
> As shown below (or see Table 31), due to using pseudo labels and the model predictions for view selection, our training time is higher than MVSelect and Random, and at similar levels to others. Yet, our test speed is similar to baselines and faster than MVSelect since no extra reinforcement learning (RL) network is needed. Moreover, MVSelect can't be applied to a new scene because of the RL framework, and it requires all the labels for training the RL network. But our methods can be applied to cross-scene tasks (see Table 2), demonstrating their generality.
>
> | Method | Memory(GB) | FLOPs(G) | Train(s) | Test(s) |
> |--------|------------|----------|----------|---------|
> | MVSelect | 16.879 | 532.200 | 1200 | 10 |
> | Random | 17.594 | 530.703 | 1700 | 7 |
> | Random (Pseudo) | 17.594 | 530.703 | 7600 | 7 |
> | IVS\_$S_g$ (Ours) | 17.594 | 530.703 | 8006 | 7 |
> | AVS\_$S_{mask}$ (Ours) | 17.594 | 530.703 | 9172 | 7 |
> | AVS\_$S_{density}$ (Ours) | 17.594 | 530.703 | 9194 | 7 |
>
> ### **Q5: Cross-domain evaluation.**
>
> MVSelect requires GT annotations of all views, making it impractical for selecting among large numbers of views to save annotation budget, and has a weak generalization ability due to the RL framework, making it not applicable to novel scenes. But our method can be performed in cross-scene settings compared to MVSelect (see Table 2). Moreover, our method can achieve good generalization with different selected view numbers or frame numbers (see Table 5, 12) with limited labels, demonstrating the idea’s generality.
>
> ### **Q6: Comparison method with pseudo labels.**
>
> Thank you for your suggestion. We conduct pseudo label for comparison methods, as shown below (or see Table 21). Compared rows 1 and 2 (or rows 3 and 4), the results with a pseudo label are better than without one. However, compared with our methods, our methods' results are still better, demonstrating the advantage of our methods.
>
> | Row | Method | MAE $\downarrow$ | MSE $\downarrow$ | NAE $\downarrow$ |
> |-----|--------|------------------|------------------|------------------|
> | 1 | CVCS (Random Selection) | 39.18 | 44.92 | 0.289 |
> | 2 | CVCS (Random Selection, Pseudo label) | 30.41 | 36.43 | 0.223 |
> | 3 | MVMS (Random Selection) | 36.65 | 43.03 | 0.271 |
> | 4 | MVMS (Random Selection, Pseudo label) | 29.94 | 35.85 | 0.221 |
> | 5 | IVS\_$S_g$ (Ours) | 14.98 | 18.93 | 0.111 |
> | 6 | AVS\_$S_{mask}$ (Ours) | 12.53 | 15.33 | 0.093 |
> | 7 | AVS\_$S_{density}$ (Ours) | **10.99** | **13.57** | **0.083** |

---

### Official Review · Reviewer_hHyY · 2025-10-31

**Soundness:** 2
**Presentation:** 1
**Contribution:** 3
**Rating:** 2
**Confidence:** 4

**Summary:**

The paper proposes two view-selection strategies for multi-view crowd counting and localization: IVS and AVS. IVS is geometry-only and greedily selects K views by maximizing a score that combines scene coverage, average distance to cameras, and view diversity. This baseline does view selection, labeling, and downstream model training as separate steps. AVS modifies the same score but replaces the geometric visible region with masks or densities predicted by the current counting/localization model. During training, AVS alternates: select a new view using model-informed scores, label it, retrain the model, and repeat until K views are labeled. Pseudo labels from unselected views are also added to improve cross-scene generalization.

**Strengths:**

- Extensive ablation experiments are provided.

- The core idea is motivated and original.

- The method targets a practical labeling/annotation bottleneck and proposes an effective solution.

**Weaknesses:**

- The introduction starts abruptly with (“To deal with severe occlusions…”) before clearly stating the task. It should first define multi-view crowd counting and localization.

- The methods section lacks a concise notation summary and problem setup. A short subsection at the start should define the mathematical symbols used, and outline the indices or sets involved in the problem setup. This will make later derivations and formulas much easier to follow and reduce ambiguity.

- The method section is not explained clearly and is hard to follow. For example, in the view-addition subsection, several elements are introduced (“selected frames,” the current view set $V_{select}$, the score $S_g$, the final target $K$, and $S = S_g$ with $N = \emptyset$) before the basic idea is stated. This forces the reader to pause and disentangle the procedure.

**Questions:**

- How robust is AVS to early model errors? If the first density maps are bad or miss a region, does the selector get “stuck” where it never chooses views that would have fixed that mistake?

- Why is the final score strictly multiplicative ($S_{sc}$ · $S_{ad}$ · $S_{vd}$) instead of a weighted sum, given that one weak term can zero out an otherwise great view?

- In the counting experiments, the stopping rule is “MAE threshold of 20” What happens if the model never reaches MAE 20 for a hard scene? Do they still add the next view or stop early?

---

> ### Author Response · Authors · 2025-11-29
> **Response to Reviewer hHyY**
>
> Thank you for your valuable suggestions. Each question is answered as follows. We hope the response has addressed your concerns. We're glad to have further communication with you. The revision in the paper is marked in blue.
>
> ### **Q1: The introduction starts abruptly with (“To deal with severe occlusions…”) before clearly stating the task. It should first define multi-view crowd counting and localization.**
>
> Thank you for your suggestion. We have improved the beginning of Section 1 by defining the multi-view crowd counting and localization tasks first.
>
> ### **Q2: The methods section lacks a concise notation summary and problem setup. A short subsection at the start should define the mathematical symbols used, and outline the indices or sets involved in the problem setup. This will make later derivations and formulas much easier to follow and reduce ambiguity.**
>
> Thank you for your valuable suggestions. We have added a notation summary table in Section 3 with indices for friendly symbol query, as shown below (or see Table 1 with page numbers). A brief illustration is provided in the preamble of Section 3 to describe the problem setup.
>
> | Symbol | Meaning | Symbol | Meaning |
> |--------|---------|--------|---------|
> | $F$ | The number of selected frames | $H_i$ | Visible scene region by view $i$ |
> | $K$ | The number of selected views | $h$ | Scene Height |
> | $v_{max}$ | The view with the largest FOV | $w$ | Scene Width |
> | $V_{select}$ | Selected view group | $D_p$ | Sum of inverse distance |
> | $S$ | View selection score equation | $S_{sc}$ | $S$ with Scene coverage |
> | $S_g$ | $S$ with view/scene geometries | $S_{ad}$ | $S$ with Average distance |
> | $S_{mask}$ | Mask-indicated $S$ | $S_{vd}$ | $S$ with view diversity |
> | $S_{density}$ | Density-indicated $S$ | $B_{k}$ | Crowd density map mask |
> | $N$ | Downstream task model | $M_k$ | Crowd density map |
> | $\emptyset$ | No $N$ | $D^{den}_p$ | $D_p$ with $M_k$ |
> | $H_s$ | Scene region | $V^k_{select}$ | $k$ selected views |
> | $H^k_v$ | Visible scene region by $V_{select}$ | $V^g$ | View set of scene's $g$ |
> | $G$ | Scene set | $E$ | the number of training epoch |
> | $V^g_{select}$ | Selected views of scene $g$ |  |  |
>
> ### **Q3: The method section is not explained clearly and is hard to follow.**
>
> Thank you very much for pointing that out. We have revised some descriptions in Section 3.1.2 to enhance the readability of our methods.
>
> ### **Q4: How robust is AVS to early model errors? If the first density maps are bad or miss a region, does the selector get “stuck” where it never chooses views that would have fixed that mistake?**
>
>
> Thank you for your question. AVS relies on the density map predicted by the model during training. Hence, we use a threshold as a hyperparameter to determine when to obtain a well-trained model for AVS view selection, thereby alleviating prediction errors caused by a poorly trained model. The ablation results for the threshold are shown in Table 15 and 32, demonstrating the efficiency of our methods.
>
> However, the view selection equation in AVS considers both the view/scene geometries and the crowds’ density level and location information, not just predicted density maps. Moreover, AVS jointly optimizes view selection and downstream task models, further alleviating errors from the predicted density map and ensuring that selected views are adapted to downstream models’ scene-level prediction. Extensive experiments (Table 2, 7) on scene-level multi-view crowd counting and localization demonstrate the efficiency of AVS.
>
> ### **Q5: The score equation uses a multiplicative instead of a weighted sum.**
>
> Thank you very much for pointing that out. We initially consider the weighted sum as our equation. However, due to the different scales among these terms, we use multiplication to calculate score. Moreover, AVS integrates view selection priors and feedback from downstream models, jointly optimizing view selection and model training to yield selected views suitable for the downstream task and improved performance. Experiments (see Table 3 and 11) on both counting and localization tasks prove the advantages of our methods.
>
> ### **Q6: The stopping rule about MAE threshold.**
>
> Thank you for your question. The MAE threshold is a hyperparameter to control when to select the next view. If the model never reaches the MAE threshold, the training process will continue with the current view number until the maximum epoch is reached. In the AVS framework, a better model can produce a more accurate density map, thereby reducing prediction errors during view selection. Consequently, we need an appropriate model to select views by controlling the MAE threshold. We conduct ablation on the threshold, as shown in Table 15, indicating that threshold=20 is more suitable and yields better performance. We also conduct experiments on the threshold for localization tasks, as shown in Table 32.

---

### Official Review · Reviewer_88Ad · 2025-11-01

**Soundness:** 3
**Presentation:** 3
**Contribution:** 3
**Rating:** 6
**Confidence:** 3

**Summary:**

The paper studies scene-level performance for multi-view crowd counting and localization under limited labeling budgets. It introduces:

• IVS (Independent View Selection): a geometry-based score combining scene coverage, average inverse person–camera distance, and view diversity to greedily pick K views (Eqs. 1–5; Fig. 2–3).

• AVS (Active View Selection): a joint selection–training loop that augments the score with model-aware terms using a binary mask and density-weighted contributions (Eq. 6–7), and leverages masked pseudo-labels to exploit unlabeled views (Figs. 6–9; Algs. 2–3).

On CVCS (multi-scene counting), AVS improves MAE to 10.99 with strong coverage (Table 1, Fig. 4). On MultiviewX/Wildtrack (single-scene localization), AVS (with only 3 labeled views) is competitive with or better than prior methods and outperforms MVSelect despite MVSelect using full labels (Table 5).

**Strengths:**

1.	Timely, practical problem: focuses on scene-level evaluation and which cameras to label, which addresses a gap in prior multi-view works.
2.	Interpretable selection criteria with clear ablations for all terms in $S_g/S_{\text{density}}$ (Tables 2 & 9; Fig. 5).
3.	Effective pseudo-label design tied to geometry/FOV masks; measured gains when used during selection and final training (Tables 4 & 6; Figs. 8–9).
4.	Cross-scene counting evidence: CVCS trains on 23 scenes and tests on 8 (multi-scene generalization).
5.	Thorough experimental suite: sensitivity to K, F, thresholds, $\lambda$, compute/time (Tables 3, 11–16, 25).

**Weaknesses:**

1. Claimed generalization to “novel new scenes” is only partially supported

>	•	The contributions explicitly state: “Our method can apply to novel new scenes, with wider application scenarios…” (bullet 3).  ￼
	•	Support exists for counting on CVCS (held-out scenes).
	•	But not for localization: MultiviewX and Wildtrack are single-scene datasets; training/testing are within the same scene (no cross-scene split).
	•	The paper also asserts MVSelect “has a weak generalization ability… not applicable to novel scenes,” yet does not run cross-scene tests for localization to back the novel-scene claim symmetrically.


2.	Related work: strengthen the single-image (“single-shot”) context and connections

> Add a short paragraph explicitly bridging these single-image methods to your frame initialization (which uses DM-Count) and to the label-efficiency narrative, clarifying what is reused vs. what is novel in the multi-view setting.

3.	Section “3” preamble is redundant

> The opening paragraph of section 3 re-explains goals/assumptions already covered in section 1–2 (focus on scene-level; limited labels; calibrated cameras; budgets F, K).

4.	Use of DM-Count but no DM-Count baseline in tables

>	•	DM-Count is used during initialization to pick frames (and specified again in App. A.2 as trained on NWPU).
	•	However, no table reports a DM-Count baseline (e.g., single-view density + simple fusion) in the counting comparisons (Table 1 / Table 7).
• Consider adding a baseline that replaces the multi-view model with DM-Count (e.g., per-view density projected to ground plane then fused), to quantify the benefit of multi-view aggregation beyond single-image capacity. At minimum, include an ablation replacing DM-Count with a trivial frame-selector to show your gains are not an artifact of the external model.

**Questions:**

1. DM-Count dependency: How much do results change if DM-Count is replaced with (i) uniform frame sampling, (ii) a lighter single-image proxy? (Only Table 16 partially addresses this.)

---

> ### Author Response · Authors · 2025-11-29
> **Response to Reviewer 88Ad**
>
> Thank you for your valuable suggestions. Each question is answered as follows. We hope the response has addressed your concerns. We're glad to have further communication with you. The revision in the paper is marked in blue.
>
> ### **Q1: Cross-scene experiments for multi-view localization.**
>
> CVCS is a multi-scene dataset, whose training set and testing set contain different scenes (different camera views as well, see Lines 353-357). For multi-view localization, we follow the SOTA comparisons and mainly conduct experiments on single-scene datasets Wildtrack and MultiviewX. As requested, we also conduct the cross-scene experiment on the CVCS dataset for the multi-view localization task with a similar setting of counting including model, $K=5$, and $F=20$. As shown below, the results with our view selection method are better than those with other view selections (MODA and F1 score are the main metrics for overall performance), further demonstrating our method's advantages.
>
> | Method | MODA↑ | MODP↑ | Precision↑ | Recall↑ | F1_score↑ |
> |--------|-------|-------|------------|---------|-----------|
> | Random | 15.4 | 58.7 | 72.4 | 24.9 | 37.1 |
> | Uniform | 24.2 | 61.0 | 80.9 | 31.6 | 45.5 |
> | Uniform (Pseudo) | 27.6 | **62.7** | **83.6** | 34.4 | 48.7 |
> | IVS | 27.8 | 62.1 | 81.6 | 36.0 | 50.0 |
> |AVS\_$S_{density}$ | **29.4** | 62.4 | 82.8 | **37.1** | **51.3** |
>
> ### **Q2, Q3: Remove redundant description and introduce the connection between single-image methods and the proposed view selection framework.**
>
> Thank you for your valuable advice. We have added a short description in related work to strengthen the relationship between single-image methods for the multi-view model and the proposed methods. Moreover, we simplify the preamble of Section "3" to reduce redundancy and enhance the description in Section "3.1.1" to clarify the single-view model's function for our method.
>
> During frame initialization, we need to choose the first frame. First, the view with the largest field of view (FOV) on the ground is used as the reference view, providing the most scene information. Then we select the frame along the reference view with the maximum crowd as the first selected frame. Hence, a pretrained single-image counting model can provide semantic information to perform frame initialization effectively.
>
> ### **Q4, Q5: DM-Count baseline, lighter single-image proxy, uniform frame sampling,  and DM-Count dependency.**
>
> Thank you for your question. All the results are shown below.
>
> | Row | Method | MAE↓ | MSE↓ | NAE↓ |
> |-----|--------|------|------|------|
> | 1 | Multi-view Model (DMCount) | 20.38 | 26.43 | 0.155 |
> | 2 | Lighter frame selector (MCNN) | 15.25 | 18.68 | 0.112 |
> | 3 | Uniform/Random frame sampling | 12.28 | 15.46 | 0.092 |
> | 4 | IVS\_$S_g$ (Ours) | 14.98 | 18.93 | 0.111 |
> | 5 | AVS\_$S_{density}$ (Ours) | **10.99** | **13.57** | **0.083** |
>
> For the frame selector, we conduct an ablation to replace DM-Count with a random frame sampling under AVS setting. As shown in rows 3 and 5 (or see Table 18), the results with DM-Count are better, indicating that a well-designed frame selector is valid compared to random frame sampling.
>
> For uniform frame sampling, because unordered frames are used, random frame sampling is equivalent to uniform frame sampling on CVCS dataset. As shown in rows 3 and 5 (or see Table 18), our method selects frames with various lighting conditions using cosine similarity, which can enhance the sample diversity and the network's robustness.
> Moreover, we also provide experiments on uniform frame sampling on Wildtrack and Multiviewx shown  below (see Table 29), where IDdiff consists of the selected frame ID mean and standard deviation, indicating that a well-designed frame sampling is better than random frame sampling.
>
> | Dataset | Wildtrack | | | MultiviewX | | |
> |---------|-----------|-----|------------|-----|----|--------|
> | Frame Select | MODA | F1 | IDdiff | MODA | F1 | IDdiff |
> | Random | 77.5 | 87.8 | 24.7±20.7 | 80.7 | 89.6 | 4.9±4.1 |
> | Uniform | 77.0 | 87.5 | 25.0±0.0 | 80.8 | 89.6 | 5.0±0.0 |
> | AVS\_$S_{density}$(Ours) | **79.2** | **88.9** | 18.0±52.3 | **81.2** | **90.0** | 5.0±6.0 |
>
> For DM-Count as the multi-view model baseline, we conduct the baseline with IVS setting, which replaces the multi-view model with DM-Count and simply fuses the projected multi-view features by maximum. Compared to rows 1 and 4 (or see Table 19), the results indicate that the multi-view architecture outperforms a single-view model with a simple fusion strategy.
>
> For a lighter single-image proxy, we also experimented with a lighter model, MCNN, to select frames using the IVS setting. The results, compared with rows 1, 2, and 4 (or see Table 19), indicate that the multi-view model is vital, but the different single-image proxy frame selector has little effect on performance.

---

### Author Response · Authors · 2025-12-03
**Final Comment for Paper #10693**

Dear Area Chair and Reviewers,

We sincerely thank you for your thoughtful feedback and valuable suggestions, which have greatly helped us improve the quality of our manuscript.

We would like to highlight that we have carefully addressed all concerns raised by the reviewers in our revised submission. Specifically, we conduct extra counting experiments on a new real-world dataset and localization experiments on the cross-scene dataset CVCS. More sufficient comparison experiments are also conducted, including methods with uniform view selection, different frame selectors, and the proposed pseudo-label. Ablation studies about the cover rate and performance of random view selection are presented. Moreover, a summary table of notations and the visualization of the camera outline are added. Refinements to the introduction, methods, and related work are made. These revisions have significantly enhanced the paper’s clarity, methodological rigor, and overall contribution to the field.

In this paper, we focus on the scene-level multi-view crowd counting and localization with limited labels. Compared to existing methods, which focus only on model performance, we propose to train the model and select views to achieve good scene-level performance under a limited label budget. Consequently, we first propose independent view selection (IVS) that considers view and scene geometries. Then, based on IVS, we propose the active view selection method (AVS), which considers downstream model predictions in view selection and jointly optimizes view selection and downstream tasks.

We fully understand the constraints imposed by the current discussion policy, which limits further interaction with reviewers on OpenReview during the rebuttal phase. Nonetheless, we believe that the revised version now meets the high standards expected for publication at ICLR 2026.

Thank you once again for your time, careful consideration, and constructive engagement with our work.

Best regards,

The Authors of Paper #10693

---

### Meta-Review · Area_Chair_dTPv · 2026-01-05

**Summary:**

This paper tackles a genuinely interesting and practical problem: how to intelligently select camera views for crowd counting and localization when labeling budgets are tight. To solve this, the authors propose two frameworks—Independent View Selection (IVS) and Active View Selection (AVS)—which move beyond random selection by utilizing geometric priors and, in the case of AVS, feedback from the downstream model itself.
The core debate during the review process centered on the cost of this intelligence. While the goal is label efficiency, the reviewers questioned whether the complex, iterative machinery of AVS was actually necessary, or if simpler, heuristic baselines could achieve similar results without the heavy computational lift.

**Reviewer Concerns:**

Several reviewers (LBIA, YB4N) rightly pointed out that comparing only against "Random" selection was unfair. The authors successfully added "Uniform" selection baselines to the discussion.

Reviewer 88Ad was uncertain about the localization claims on single-scene datasets. The authors effectively countered this by running new cross-scene localization experiments on the CVCS dataset, showing their method can generalize across scenes.

There are remaining concerns.

This is the most critical issue. Reviewer LBIA highlighted that the iterative training loop makes AVS 6-7 times slower to train than existing methods like MVSelect. While the authors argue that test-time inference is fast, an "efficient learning" method that requires an order of magnitude more training time is a difficult sell.

The new "Uniform" baseline results were revealing. While AVS is statistically better, the gap is often quite narrow. For example, on the CityStreet dataset, the simple Uniform baseline achieved an MAE of 11.82, while the complex AVS method achieved 9.80. Reviewers remained unconvinced that a 7x increase in training cost is justified by these incremental gains, especially when the "CoverRate" metric shows a negligible difference between strategies

Reviewer hHyY pointed out that the core scoring function feels ad-hoc. The authors confirmed in the rebuttal that they chose a multiplicative combination of scores simply because a weighted sum struggled with scale differences10. This lack of theoretical grounding, combined with the reliance on sensitive hyperparameters like thresholds ($\tau$) to prevent the model from learning errors11, leaves the impression of a fragile system that requires careful tuning

**Reviewer Scores:**

Reviewer 88Ad  6.

Reviewer YB4N: 4.

Reviewer hHyY: 2.  While the paper reads better now, the major concern about the heuristic nature of the solution and its robustness remains unresolved.

Reviewer LBIA: 2. The rebuttal data actually reinforced this reviewer's negative stance; by providing the training times and Uniform baselines, the authors confirmed that the method is computationally expensive.

---

### Decision · Program_Chairs · 2026-01-26

Reject